

# Towards the Marine Arctic Component of the Pan-Eurasian Experiment

Timo Vihma[1], Petteri Uotila[2], Stein Sandven[3], Dmitry Pozdnyakov[4], Alexander Makshtas[5], Alexander Pelyasov[6], Roberta Pirazzini[1], Finn Danielsen[7], Sergey Chalov[8], Hanna K. Lappalainen[2,9], Vladimir Ivanov[5,8,10], Ivan Frolov[5], Anna Albin[7], Bin Cheng[1], Sergey Dobrolyubov[8], Viktor Arkhipkin[8], Stanislav Myslenkov[8], Tuukka Petäjä[2,9], Markku Kulmala[2,9]

[1]Finnish Meteorological Institute, Helsinki, Finland
[2]Institute for Atmospheric and Earth System Research/Physics, University of Helsinki, Finland
[3]Nansen Environmental and Remote Sensing Centre, Bergen, Norway
[4]Nansen International Environmental and Remote Sensing Centre, St. Petersburg, Russia
[5]Arctic and Antarctic Research Institute, St. Petersburg, Russia
[6]Center for the Arctic and Northern Economies, Council for Research for Productive Forces, Moscow
[7]Nordic Foundation for Development and Ecology, Copenhagen, Denmark
[8]Moscow State University, Moscow, Russia
[9]Tyumen State University, Tyumen, Russia
[10]Hydrometeorological Center of Russian Federation, Moscow, Russia

*Correspondence to*: Timo Vihma (timo.vihma@fmi.fi)

**Abstract.** The Arctic marine climate system is changing rapidly, seen as warming of the ocean and atmosphere, decline of sea ice cover, increase in river discharge, acidification of the ocean, and changes in marine ecosystems. Socio-economic activities in the coastal and marine Arctic are simultaneously changing. This calls for establishment of a marine Arctic component of the Pan-Eurasian Experiment (MA-PEEX). There is a need for more in-situ observations on the marine atmosphere, sea ice, and ocean, but increasing the amount of such observations is a pronounced technological and logistical challenge. The SMEAR (Station Measuring Ecosystem-Atmosphere Relations) concept can be applied in coastal and archipelago stations, but in the Arctic Ocean it will probably be more cost-effective to further develop a strongly distributed marine observation network based on autonomous buoys, moorings, Autonomous Underwater Vehicles (AUV), and Unmanned Aerial Vehicles (UAV). These have to be supported by research vessel and aircraft campaigns, as well as various coastal observations, including community-based ones. Major manned drifting stations may occasionally serve comparable to terrestrial SMEAR Flagship stations. To best utilize the observations, atmosphere-ocean reanalyses need to be further developed. To well integrate MA-PEEX with the existing terrestrial/atmospheric PEEX, focus is needed on the river discharge and associated fluxes, coastal processes, as well as atmospheric transports in and out of the marine Arctic. More observations and research are also needed on the specific socio-economic challenges and opportunities in the marine and coastal Arctic, and on their interaction with changes in the climate and environmental system. MA-PEEX will promote international collaboration, sustainable marine



meteorological, sea ice, and oceanographic observations, advanced data management, and multidisciplinary research on the marine Arctic and its interaction with the Eurasian continent.

# 1 Introduction

Since the 1990s the Arctic climate has warmed at least twice as fast as the global mean (Francis and Vavrus, 2015) with

associated dramatic declines in sea ice cover (Kwok and Cunnigham, 2015), terrestrial snow pack (Derksen et al., 2015) and permafrost (Lawrence et al., 2015), coincident with increase in continental freshwater discharge into the Arctic Ocean (Carmack et al., 2016) and changes in the biochemical cycle (Shakhova et al. 2007; Harada, 2016). These changes have strong socio-economic and environmental impacts. Global economic interest in the Arctic has strongly increased. In particular, decrease of sea ice along the Northern Sea Route (NSR) will allow intensifying navigation, which is already occurring in the

western parts of the route (Liu and and Kronbak, 2010). Although transit navigation through the entire NSR is still very limited and restricted to a short season in late summer – early autumn, there is a growing interest towards more extensive transit navigation (Smith and Stephenson, 2013). This interest and associated increase in Arctic research and technology development have been particularly strong among Asian countries: China, Japan and South-Korea. The Chinese initiative One Belt One Road (Tsui et al., 2017) together with the Chinese-Russian Ice Silk Road (Sørensen and Klimenko, 2017) are and will be

facilitating the ongoing economic changes in the Arctic regions. In addition to navigation, economic interest towards the Arctic Ocean and its marginal seas is also growing due to the large off-shore hydrocarbon resources, fisheries, and tourism (AMAP, 2017).

The rapid climate change and increasing industrial and transport activities generate large risks for the sensitive Arctic environment. Both climate warming and increasing economic activities will have environmental impacts in the marine Arctic.

Major environmental impacts related to climate warming include ocean acidification (AMAP, 2013), changes in the availability of nutrients, and numerous changes in marine ecosystems, e.g. in primary production and productivity, phytoplankton biomass and species composition, and fish species diversity (AMAP, 2017). The environmental impacts of increasing economic activities include worsening of air and water quality. Even more alerting than gradual trends is the increasing risk of accidents that may result in major oil spills.

Increases in navigation, other offshore activities, aviation, and tourism as well as the associated increasing risk of environmental hazards necessitate more accurate and extensive operational forecasts for weather, sea ice, and open ocean conditions in the Arctic. To respond to this need, the Copernicus Marine and Atmospheric Services have since 2014 provided monitoring and short-term forecasting on global scale, including the Arctic (von Schuckmann et al., 2016). The services use various models, satellite data and available in situ data that are delivered in near real-time. However, the quality of the

Copernicus services in the Arctic is uncertain, and one of the reasons is the lack of in situ data. Above all, more data on atmospheric pressure, wind, temperature, and humidity need to be collected to be assimilated to numerical weather prediction



(NWP) models (Inoue et al., 2013). Data on sea ice concentration and thickness as well as sea surface temperature and other ocean variables are needed for operational sea ice and NWP models.

Further, changes in the Arctic may also have extensive impacts on non-Arctic regions, related at least to weather and climate, in particular in central and east Asia (Uotila et al., 2014; Kug et al., 2015; Overland et al., 2016) as well as on global

economics, at least in the transport (Furuichi and Otsuka, 2013) and hydrocarbon sectors (McGlade, 2012) Hence, it is a global-scale challenge and need for the societies to ensure the sustainable development and future of the Arctic maritime environments. Among the first practical steps are identification of processes of a high research priority and establishment of a coherent, coordinated, comprehensive observation system.

Quantification of the state, variability and trends in the Arctic climate system is hampered by both the sparsity of in-situ

observations from the marine Arctic and challenges in retrieval of useful data from satellite remote sensing in the Arctic. The latter challenges are related, among others, to high latitudes (no data from geostationary satellites), the long Polar night (no data from optical sensors), and the ice and snow surface (difficulties to distinguish between signals originating from atmospheric water/ice and the surface). Despite of these challenges, satellite remote sensing is a vital component of the Arctic observation system. Especially, the Sentinel programme under Copernicus, providing open and free access to satellite data,

has dramatically increased the production and distribution of satellite data from Arctic regions (https://scihub.copernicus.eu/). In-situ and surface-based remote sensing observations are needed for a large number of variables that cannot be satisfactorily observed from satellites. In general, part of the in-situ and surface-based remote sensing observations are taken during research cruises, manned ice stations, and research aircraft campaigns, and part are collected applying longer-lasting installations. The latter include drifting buoys on sea ice and ocean surface; ice-tethered platforms, gliders and floats for subsurface observations;

moored chains with oceanographical instruments; and coastal stations. There are more ocean observations available from the marginal seas (Barents, Laptev, Bering, Greenland, and Labrador seas) than from the central Arctic, and more from summer than winter. Observational information on sea ice concentration and extent are satisfactory since the availability of passive microwave satellite remote sensing data in 1979, but information on sea ice thickness is based on sparse submarine data from several decades before 2000 and aircraft observations combined with in- situ and satellite remote sensing data during the last

two decades (Kwok and Cunningham, 2015). In-situ observations on the Arctic atmosphere are restricted to research vessel and aircraft campaigns (mostly in summertime), occasional manned drifting ice stations, regular coastal observations, as well as sea-level pressure and air temperature observations applying buoys deployed on sea ice.

The Pan-Eurasian Experiment (PEEX) program is interested in contributing to a comprehensive in situ observation system for the marine Arctic based on a combination of distributed, mostly autonomous, observations and flagship stations following

the SMEAR (Station Measuring Ecosystem-Atmosphere Relations) concept, successfully applied in the Eurasian continent. The system is to be designed in collaboration with other programs addressing the present and future observation networks in the Arctic, including the Sustaining Arctic Observation Networks (SAON) and the Arctic Monitoring and Assessment Program (AMAP) established by the Arctic Council, the European Commission project Integrated Arctic Observation System



(INTAROS), and several other programmes and networks (Section 3.1). The regional focus of PEEX has so far been in the Eurasian continent. In the PEEX approach, large-scale research topics are studied from a system perspective to fill the key gaps in our understanding of the interactions and feedbacks in the land–atmosphere–aquatic–society continuum (Lappalainen et al., 2014; 2016; 2018). So far PEEX has had a hydrological component addressing terrestrial waters but not a marine

component. Due to the importance of the marine Arctic in the climate system and the increased economic interest in the Arctic regions, it is vital that PEEX includes an active marine component, addressing physical and ecosystem processes in the ocean, sea ice, and marine atmosphere, and their alterations due to climate and environmental drivers.

The objective of this paper is to design the Marine Arctic Component of PEEX (MA-PEEX), schematically illustrated in Figure 1. This requires evaluation of the state and change of the marine Arctic climate and environmental system (Section 2),

identification of the actual research needs and the state of existing observations in relation to the needs (Section 3), evaluation of the information available on the basis of atmospheric and ocean reanalyses (Section 4), evaluation of the relevant socio-economic aspects that both affect and are affected by climate and environmental changes (Section 5), and assessment of the challenges and emerging opportunities in building MA-PEEX (Section 6). It is aimed to be integrated with the well-established structure and activities of the terrestrial and atmospheric components of PEEX. This requires particular attention to the linkage

and feedback processes, such as atmospheric transports in and out of the Arctic, river discharge, and various other coastal processes.

## 2 The Arctic climate and environmental system

### 2.1 Arctic marine climate system

The Arctic Ocean and its sea ice cover have a major role in the regional and global climate system. Central components of this

role are the large heat capacity and freshwater storage of the ocean (Carmack et al., 2016) as well as the properties of the sea ice and its snow pack to both insulate (during most of the year) the relatively warm ocean from the cold atmosphere and to reflect the major part of the incoming solar radiation back to space. The ocean heat capacity per unit volume is over three orders of magnitude larger than that of the overlaying atmosphere, which keeps temperatures much more constant in the ocean than atmosphere. The surface albedo of sea ice ranges from 0.4 for melting ice to 0.85 for ice covered by dry, new snow

(Perovich and Polashenski, 2012), compared to less than 0.1 for the open ocean. The heat conductivity of sea ice and, in particular, snow is very low, which allows the maintenance of a very large vertical temperature difference between the ice base (at the freezing point) and the snow surface (in winter as cold as -50°C. Arctic Sea ice is typically 1-3 m thick having a smaller heat capacity than the other components of the climate system, which makes it a highly sensitive indicator of climate variability and change. The sea ice cover also strongly reduces the fluxes of momentum (Leppäranta, 2011), heat and moisture (Lüpkes

et al., 2008), $CO_2$ (Parmentier et al., 2013), $CH_4$ (Shakhova et al., 2015) and other gases between the atmosphere and the ocean.



The Arctic atmosphere receives a lot of heat and moisture via horizontal advection from lower latitudes and, over ice-free areas, via vertical turbulent fluxes from the ocean. Over the ice-covered parts north of 70oN, the transport of dry static energy and latent heat from lower latitudes is approximately balanced by the longwave radiative cooling (Persson and Vihma, 2017), but allows much higher air temperatures than would occur without the transport from lower latitudes. The Arctic atmosphere

contains water in the forms of vapour, liquid, and ice, the total water content being approximately 200 km$^3$ (Serreze et al., 2006). This is a minor volume compared to the water content of the Arctic Ocean, ice sheets, glaciers, lakes, rivers, and ground. However, the atmospheric water is of fundamental importance for the Arctic water cycle: the freshwater residence is approximately a week in the Arctic atmosphere (Serreze et al., 2006), compared to a decade in the Arctic Ocean (Carmack et al., 2016) and thousands of years in ice sheets and glaciers. The atmospheric boundary layer (ABL), mainly stable stratified,

plays an important role as a buffer between the Earth surface and the bulk of the atmosphere. Hence, understanding the physics of the ABL is a prerequisite for reliable modelling of the present and future climate in the Arctic (Vihma et al., 2014).

During the recent decades the Arctic air temperatures have increased at least twice as fast as the global mean (Francis and Vavrus, 2015). This is called the Arctic amplification of climate warming. The warming has been strongest in winter with the maxima over sea ice, whereas in summer the warming has been weaker with the maxima in terrestrial Arctic and Greenland

ice sheet (Figure 2). The atmospheric warming is associated with strong declines in sea ice: since the early 1980s, the September sea ice extent had decreased by approximately 40% and the cold season sea ice thickness by approximately 50% (Kwok and Cunnigham, 2015). Since 1950s, the decrease in sea ice area has almost linearly followed the increase of the cumulative CO2 emissions (Notz and Stroeve, 2016).

Atmospheric moisture, clouds, and precipitation simultaneously affect and are affected by the recent rapid climate change

in the Arctic. Over most of the year, clouds heat the sea ice surface via enhanced downward longwave radiation (Shupe and Intrieri, 2004), and precipitation brings freshwater to the Arctic Ocean both directly (exceeding evaporation) and, above all, via river discharge from the surrounding continents. These high freshwater fluxes are the main reason for a lower salinity in the Arctic Ocean compared to other oceans, and are crucial for the stratification, sea ice, chemical budget, and circulation patterns of the Arctic Ocean (Rudels, 2012; Polyakov et al., 2008). Due to increased precipitation, the river discharge has

increased by approximately 30% in the period 2000-2010 (Haine et al., 2015) compared to 1979-2001 (Serreze et al., 2006). The precipitation increase is projected to continue during this century (Vihma et al., 2016). In the PEEX study region, the river discharge to the Arctic originates from Greenland, northern Scandinavia, north-eastern parts of the European plain, Polar Ural, Altai and Sayany mountains, West and East Siberia, East Kazakstan, Mongolia, and China.

**1.2 Arctic marine ecosystems**

Climate warming in the Arctic is recognized as a powerful driver of significant changes in the Arctic Ocean primary production, PP (Petrenko et al., 2013). PP in the entire ice-free Arctic basin has increased by ~16% over 13 years (1998–2010), which is primarily a result of the drastic sea ice decline, but also due to the continuous growth of phytoplankton annual




productivity, which has been approximately 32% higher than during 1959-2005 (Petrenko et al., 2013). In the marginal zone of the Arctic Ocean the PP has increased less primarily due to the influence of river-runoff increase, ensuing water turbidity and worsening of water quality (Pozdnyakov et al., 2007). The higher gross primary production (GPP) would affect air-sea fluxes of $CO_2$. Also an increase in the overall biological production including the production of higher trophic level organisms

and fish populations could be foreseen (Doney et al., 2012). The warmer surface waters may enable the invasion of new species, which may dramatically impact the sensitive Arctic ecosystem by changing the pelagic food webs, energy flows and biodiversity. This aspect is very relevant for the regulation of international fisheries in the Arctic.

The melting of permafrost together with increasing precipitation in the Arctic river basins may lead to flooding, and increasing the amount of freshwater and allochthonous materials in the Arctic shelves, and further in the Arctic Basin. All

these processes may impact the Arctic Ocean marine ecosystems, their productivity, and the key biogeochemical cycles in the region. The effects of the increased GPP and phytoplankton biomass on the productivity of higher trophic levels of the Arctic ecosystem are, however, not well known. In typical Arctic ecosystems the most important primary consumers are large-sized herbivorous copepods, which have lifecycles synchronized with the seasonal algae bloom (Kosobokova, 2012). Among the key questions is what would be the joint effect of Arctic warming, ocean freshening, pollution load and acidification on the

Arctic Ocean ecosystem (Janout et al., 2016), primary production (Vancoppenolle et al., 2013; Arrigo and van Dijken, 2015) and carbon cycle (Christensen et al., 2017). Considering the Arctic marginal seas, however, high volumes of additional GPP seem highly unlikely. This is due to strong nutrient limitations after the end of the spring bloom, the low activity of bacterioplankton (slow nutrient regeneration), low temperatures (Makkaveev et al., 2010; Sazhin et al., 2010), and worsening of the water column light climate driven by runoffs of sediment and dissolved organic-rich riverine waters (Pozdnyakov et al.,

20    2007).

The forthcoming dynamics of marine ecology changes can hardly be straightforwardly projected to the future, as the issues of availability of nutrients, bottom-up trophic interactions (especially, through the phytoplankton-zooplankton-fish levels) remain moot and require further investigations. The nutrient balance dynamics might also be a controlling factor as the freshwater-driven enhancement of stratification comes to prevail over the nutrient supply from deep waters (Olli et al. 2002;

Hansen et al. 2003). Reduction of ice cover also entails the problem of vertical turbulent mixing, in pelagic tracts of the Arctic Ocean, and a reduction of the pan-Arctic ice-edge stretch, which is the zone of enhanced PP (Syvertsen, 1991; Johnsen, 2018). The role of ice cover is pivotal in terms of controlling the light climate throughout the water column, configuring the euphotic zone, and controlling the atmospheric nitrogen fluxes into the water column (essential for blue-green algae uptake of the nitrogen from the air), determining the primary productivity in the ice cover column as a source of initiation/boosting in-water

microalgae, particularly in off-ice edge areas extending under the ice sheets (Arrigo et al., 2008).

The oil and gas drilling and navigation in the Arctic shelf areas are a major concern for marine Arctic ecosystems today and in the future. The associated impacts will have long-term effects because of the high vulnerability sensitivity of the Arctic ecosystems. On one hand, there is a risk of irreversible changes in marine Arctic productivity and key biogeochemical cycles.



On the other hand, the changes may generate potential for $CO_2$ absorption by marine ecosystem. Processes involving the Arctic may also affect adjacent boreal areas. All this calls for a systematic monitoring of environmental parameters of the Arctic Ocean and its marginal seas.

## 3 Existing observations and processes to be studied

Numerous processes are acting in the marine Arctic climate system: in the ocean, sea ice, and atmosphere. Many of these processes act on a subgrid-scale, and they accordingly need to be parameterized in Earth system models and operational NWP and sea ice models (Vihma et al., 2014). However, there is also a strong need to better understand synoptic and hemispherical-scale processes (Zhang et al., 2004; Crawford and Serreze, 2016; Overland et al., 2016). From the point of view of MA-PEEX, particularly important processes to be studied are those responsible for the interaction between the marine, atmospheric, and

terrestrial components of PEEX. Such processes include the river discharge (Bring et al., 2017) and its impacts on the sea ice and ocean, including the water column light climate (Pozdnyakov et al., 2007; Carmack et al., 2016), storm surges (Wicks and Atkinson, 2017), coastal erosion (Overduin et al., 2014), transports of heat and freshwater (Dufour et al., 2016), aerosols (Ancellet et al., 2014; Popovicheva et al, 2017),), and air pollution (Bourgeois and Bey, 2011; Law et al., 2015) from lower latitudes to the central Arctic; as well as cold-air outbreaks and planetary wave trains originating from the Arctic and affecting

Eurasian weather and climate (Vihma, 2014; Kug et al., 2015). Process understanding as well as quantification of the state, variability and trends in the Arctic climate system are hampered by the sparsity of observations from the marine Arctic. This is related to the high cost of observations, difficult accessibility to the measurement sites, and the harsh environment for instruments.

Below we first introduce the most important observation networks (Section 3.1). Then we describe in more detail the key

processes in the atmosphere, sea ice, and ocean, also evaluating how well these can be understood and quantified on the basis of existing observations (Sections 3.2 to 3.5).

### 3.1 Observation networks

Surface-based atmospheric observations in the Arctic mostly rely on land-based stations located along the continental coasts and in the main Arctic islands (such as Svalbard, Severnaya Zemlya, and Greenland). These stations belong to one or more observational networks, defined on the basis of the target observed parameters, which may have global, continental, or regional coverage. Networks with global coverage that include Arctic stations are e.g. the Global Atmosphere Watch (GAW), the Global Cryosphere Watch (GCW), the global radiosonde network (accessible via the Integrated Global Radiosonde Archive

(IGRA)), the Global Climate Observing System (GCOS) Upper-Air Network (GUAN), and the GCOS Reference Upper-Air Network (GRUAN).



Station-based networks specific for the Arctic include the International Arctic System for Observing the Atmosphere (IASOA), the Arctic Coastal Dynamics (ACD), the Arctic Hydrological Cycle Monitoring, Modelling, and Assessment Program (Arctic-HYDRA), and the IASC Network on Arctic Glaciology. The International Arctic Buoy Programme (IABP) collects surface observations from a vast Arctic marine area, but still it does not sufficiently cover the Eurasian sector (Figure 3).

Other networks covering only Greenland include the Programme for Monitoring of the Greenland Ice Sheet (PROMICE) and the Greenland Climate Network (GC-NET).

Some European Research Infrastructures (RI) that ensure long term sustainability to atmospheric and ocean observations extend their spatial coverage up to the Atlantic sector of the Arctic. These include the European Aerosols, Clouds and Trace gases RI (ACTRIS), the European contribution to the Argo program RI (Euro Argo), the Integrated Carbon Observation System RI (ICOS), the German ocean observing system "Frontiers in Arctic marine Monitoring (FRAM)", and the Danish biochemical observation network for Greenland coastal seas, the ship of opportunity network, and the tide gauge network). However, in the largest part of the marine Arctic there are no monitoring infrastructures. In general, most of the existing marine Arctic data, including both atmospheric and ocean observations, are collected under time limited research projects. It is therefore crucial to establish a sustained integrated Arctic Observing System that would foster the long-term sustainability, monitoring enhancement, and harmonization of the Arctic observations, to improve the scientific understanding of the complex and sensitive Arctic environment. This is the objective of the ongoing EU project INTAROS (Integrated Arctic Observation System). INTAROS has focus on developing integrated observing platforms that can observe ocean, sea ice and atmospheric variables from a suite of sensors. Such integrated platforms are important for both process studies and long-term monitoring. The platforms can be ice-buoys, ocean buoys or underwater systems such as bottom-moorings, gliders, floats and Autonomous Underwater Vehicles (AUVs). INTAROS will furthermore develop coordination and collaboration between data providers in the pan-Arctic region in order to better use existing systems and resources from many countries. INTAROS collaborates with other projects in the EU Arctic Cluster (http://www.eu-polarnet.eu/eu-arctic-cluster/) to ensure that research efforts are well coordinated. ARICE is one of the projects in the cluster with specific objective to increase the availability of icebreakers to enable more data collection in the Arctic Ocean. INTAROS is also building collaboration with institutions in USA, Canada, Japan, China and South Korea who are active in ocean observing in the Arctic. The goal is to establish a Pan-Arctic Ocean observing system that can deliver data from all the Arctic sub-regions.

Drifting ice stations have played a major role in the history of Arctic Ocean observations. The first in the series of the Soviet Union "North Pole" stations was operated in 1937-1938, followed by 30 stations during 1950-1991. In this century Russia continued to perform the comprehensive monitoring of the natural environment of the Central Arctic and studies of the physical processes that determine its state (Figure 4). These studies are especially important in terms of improving climate models. To obtain the new data about the above-mentioned processes, complex hydrometeorological observations had been organized at the drifting stations "North Pole-32" to "North Pole-40" in 2003 – 2014 (Figure 5). They included observations



in atmosphere, sea ice and ocean with special attention to interactive processes. The most important western drifting stations have been the Surface Heat Budget over the Arctic Ocean (SHEBA) in 1997-1998, the Tara expedition during the International Polar Year in 2007-2008, and the Norwegian N-ICE expedition in the European Marginal Ice Zone in winter 2015. The Multidisciplinary drifting Observatory for the Study of Arctic Climate (MOSAiC) will be the next major international experiment from 2019-2020, where the focus is studies of Arctic climate and ecosystem processes (http://www.mosaicobservatory.org/). These drifting stations provided unique possibilities to study snow and sea ice properties and thermodynamics, surface energy budget, structure and dynamics of the ABL, atmospheric gas composition, air-ice-ocean exchange of carbon dioxide and methane, as well as ozone observations, which for the first time instrumentally recorded the appearance of ozone hole in the Central Arctic in March 2011 (Manney et al., 2011). Drift of the stations naturally provides some spatial coverage, but also causes challenges to distinguish between temporal and spatial variations. The same challenge is present in interpretation of data from drifting buoys, subsurface floats, research vessels, and aircraft. Research cruises and aircraft campaigns have mostly taken place during summer, and in most cases rather close to the ice margin.

It was only in the early nineteenth century that river discharge was systematically monitored, first in northern Europe, most notably in the Baltic Sea catchments, and 50 years later in North America (Gulf of St. Lawrence). The number of monitoring stations for river discharge reached its maximum during the 1980s, when about 74% of the total non-glacierized Pan-Arctic was monitored (Shiklomanov and Shiklomanov 2003). Later, there was significant decline in gauges in Russia firstly due to population decreases in high-altitude areas, loss of qualified personnel, and insufficient financial support. The total Pan-Arctic area monitored decreased by 67% from 1986 through 1999, and in Russia the decrease was 79% (Shiklomanov et al. 2002). The monitoring of sediment and water quality components is a few times scarcer; approximately only 10 % of the catchment area is monitored. Due to major rivers, only 12 hydrologic gauges are sufficient to capture 91% of total monitored area and 85% of total monitored discharge. However, for a detailed description of the state of Arctic land surface hydrology, it is necessary to record the discharge from much smaller sub-basins throughout the entire Pan-Arctic landmass.

To develop "quick-look" outputs that characterize terrestrial and aerological water budgets across the pan-Arctic drainage region and to create hydrologically-based re-analysis products and to analyze these time series of spatial and temporal variability of the pan-Arctic land mass, Arctic-RIMS (Rapid Integrated Monitoring System) project is now being implemented (http://rims.unh.edu/ background.shtml). Additionally, the R-ArcticNET historical archive and operational ArcticRIMS offer an important opportunity to monitor the progressive changes of the hydrological cycle by providing an historical benchmark against which future conditions can be compared. The Global Runoff Data Centre (GRDC; http://www.grdc.bafg.de) in Koblenz, Germany, has the largest and most actively updated global database.

In the context of Arctic warming that promotes terrestrial permafrost thawing and shifting hydrologic flowpaths, leading to fluvial mobilization of sediment and associated chemicals, geochemical fluxes from Eurasian rivers into the Arctic play a role in global hydrochemical cycles. Due to mentioned scarcity in hydrological monitoring, the main datasets are based on regional studies recently performed in Lena (Hölemann et al. 2005), Ob (Shakhova et al. 2007), and Amur (Levshina 2008;



Chudaeva et al. 2011) rivers, and summarised in reviews (Savenko 2006; Bagard et al. 2011; Pokrovsky et al. 2015). Some of the large rivers (e.g. Kolyma) have never been studied in regard to sediment geochemistry. Intensive sampling was initiated by the Pan-Arctic River Transport of Nutrients, Organic Matter, and Suspended Sediments (PARTNERS) project and later continued through the Arctic Great Rivers Observatory (Arctic-GRO). These efforts have led to major revisions of flux estimates for dissolved organic matter (DOM) and inorganic nutrients (Holmes et al. 2012), as well as improved understanding of the composition of DOM delivered to Arctic coastal waters. An additional scientific problem is related to the ongoing changes in Arctic fluxes, which are impacted by the sources trends across the watershed, which transmits the hydrogeochemical signal through the river system. The PEEX subprogram Selenga-Baikal Network aims at integrated field-based and modeling knowledge of hydroclimatic and land use changes to develop conceptual framework for a distributed rainfall-runoff-sediment-heavy metal module at the catchment scale for water surface - atmosphere interactions and feedbacks (Kasimov et al. 2017).

In addition to climate and environmental research, there is an increasing need for observations from the marine Arctic to support development of operational services. Above all, more data on atmospheric pressure, wind, temperature, and humidity should be collected to be assimilated to numerical weather prediction (NWP) models. Data on sea ice concentration and thickness as well as sea surface temperatures are needed for operational sea ice, ocean, and NWP models. These needs are recognized by the international community, and one of the concrete responses is the Year of Polar Prediction (YOPP) from summer 2017 to summer 2019, which is enhancing observational and modelling activities in the Arctic. YOPP includes data assimilation experiments to quantify the benefit of various observations on operational weather, sea ice and sea state forecasts. Evaluation of the results of these experiments will help the plan the most beneficial additions to the present Arctic observation network. Further, closer interaction between the NWP model developers, forecast providers, and forecast users should include interactive elicitation of user needs, stepwise co-development of needs and capabilities, and assessment of service improvement response thresholds. Another major opportunity for MA-PEEX is the year-round drifting ice station MOSAiC scheduled from autumn 2019 to autumn 2020.

### 3.2 Marine atmosphere

The most important atmospheric processes over the marine Arctic can be divided into the following categories: (a) atmospheric boundary layer turbulence and exchange processes at the air-ice and air-water interfaces, (b) aerosol and cloud physics, (c) orographically and thermodynamically driven coastal processes, (d) synoptic-scale cyclones and Polar lows, (e) circumpolar heat and moisture budgets, (f) stratosphere-troposphere coupling, (g) local and large-scale processes affecting air quality, and (h) Arctic – midlatitude linkages affecting weather and climate.

To better understand and quantify these processes and their mutual interactions, there is a strong need for more in-situ and remote sensing observations supported by model experiments. The regular atmospheric in-situ observations from the marine Arctic are restricted to coastal stations as well as sea level pressure and air temperature observations taken by drifting buys





deployed above all by IABP. Presently (May 2018) there are almost 40 buoys on Arctic sea ice (Figure 3). The sea level pressure observations are important to detect the synoptic-scale pressure field, which is needed for climatology, meteorological research, and, above all, for initialization of NWP models (Inoue et al., 2013; 2015) and to provide atmospheric forcing for ocean and sea ice models. The air temperature observations from IABP buoys are of less good quality, as the sensors are sometimes covered by snow. More sophisticated buoys have been deployed e.g. by the North Pole Environmental Observatory (Perovich et al., 2014).

There are very few in-situ data on the vertical structure of the Arctic atmosphere. Hence, satellite remote sensing on the vertical profiles of air temperature and humidity provide an attractive source of information. However, the vertical resolution of satellite remote sensing products is too coarse for many detailed studies on the Arctic ABL, and problems remain in remote sensing of cloud water and ice contents over sea ice. In situ observations on vertical profiles are needed for more accuracy and better vertical resolution. In the marine Arctic, radiosonde soundings up to the altitudes of 15-20 km and tethersonde soundings up to 1-2 km are restricted to research cruises (Lüpkes et al., 2010; Brooks et al., 2017) and manned ice stations (Tjernström and Graversen, 2009; Vihma et al., 2008; Jakobson et al., 2013; Mielke et al., 2014). In addition, lidars, sodars, cloud radars, and scanning microwave radiometers have been used to observe the vertical profiles of wind, temperature, humidity, cloud properties, and aerosols, but such data are restricted to a few campaigns (Tjernström et al., 2012; Mielke et al., 2014).

In-situ observations on cloud properties, aerosols, air quality, as well as surface fluxes of momentum, heat, moisture, aerosols, and greenhouse gases are mostly restricted to individual field campaigns. However, regular observations at coastal and archipelago sites (e.g. Ny Ålesund, Barrow, Tiksi, and Eureka) and a newer observatory, organized in 2014 at the Cape Baranova (Makshtas and Sokolov, 2014; Uttal et al., 2016) are a remarkable source of information on atmospheric conditions over the marginal zone of the Arctic Ocean. Long-term in-situ observations have also been gathered at manned drifting stations (Section 3.1), but the spatial coverage of observations is poor, and there have been long gaps without stations.

Important providers of continuous atmospheric observations on monthly time scales have been research cruises, above all those by the German R/V Polarstern since 1980s, Swedish icebreaker Oden since 1990s, as well as Chinese and Japanese vessels more recently. Atmospheric observations taken at the drifting stations and research cruises have been crucial to research progress in the fields of vertical structure of the Arctic atmosphere (Serreze et al., 1992; Palo et al., 2017), surface exchange processes (Jordan et al., 1999; Persson et al., 2002; Andreas et al., 2010a,b), cloud physics (Tjernström et al., 2012; Shupe et al., 2013; Sedlar and Shupe, 2014), aerosols (Tjernström et al., 2014). Coastal radiosonde sounding observations have been applied in studies of meteorological processes over the ocean (Maistrova et al., 2003; Tetzlaff et al., 2013).

Research aircraft observations have been an important source of information on air-ice momentum flux and aerodynamic surface roughness (Lüpkes et al., 2013), ABL physics, in particular the evolution of stable boundary layer during on-ice flows (Vihma et al., 2003; Brümmer and Thiemann, 2001; Tisler et al., 2008) and the growth of convective boundary layer during off-ice flows (Chechin and Lüpkes, 2017), as well as mesoscale processes, such as low-level jet formation, during flows parallel to the ice margin (Langland, 1999; Guest et al., 2018). Moreover, aircraft observations have been applied to study the





radiative and microphysical properties of the Arctic clouds (Ehrlich et al., 2008; Schäfer et al., 2015), the optical characteristics of the sea ice surface (Tshudi et al., 2001), and surface-atmosphere fluxes of greenhouse gases as well as latent and sensible heat (Kohnert et al., 2014; Hartmann et al., 2018).

In-situ atmospheric observations in the marine Arctic also include several technical and environmental challenges, such as

riming of instruments, darkness of the Polar night, instability of sea ice as a measurement field (leads may open within the field, causing danger for instruments and people), tilting of weather masts due to sea ice motions, low clouds and fog hampering airborne (research aircraft, Unmanned Aerial Vehicle (UAV), and tethered balloon) operations, Polar bears' interest towards the measurement devices, disturbance of the airflow caused by ships, tents, and other constructions on ice stations (largest in conditions of stably-stratified ABL typical of the Arctic).

### 3.3 Sea ice

There are several dynamic and thermodynamic processes that need to be better observed to sufficiently quantify the state and change of the sea ice cover, and to better understand the physical mechanisms responsible for the changes. Considering sea ice thermodynamics, the key processes are (a) sea ice formation and growth, including snow accumulation on top of sea

ice and formation of superimposed ice, (b) sea ice and snow melt, including ice and snow albedo, aerosol deposition on snow and ice, and evolution of melt ponds. Possibilities to observe the spatial distribution and temporal evolution of sea ice, snow and melt ponds in the Arctic Ocean have recently improved due to better satellite remote sensing methods, airborne electromagnetic mapping methods, and sea ice mass-balance buoys. Challenges still remain, among others, in distinguishing between melt ponds and leads under cloudy skies, as well as between surface snow and clouds. Detecting ice ridges is an

observation challenge particularly important for navigation. Layers of granular ice, formed due to refreezing of flooded or partly melted snow pack, on top of columnar ice may be observed using mass-balance buoy data, the interpretation of which is supported by thermodynamic modelling (Cheng et al., 2014). Such layers may become more common due to thinning sea ice and increasing precipitation, favouring heavier snow load on top of thin ice, which increases the occurrence of flooding (Borodkin et al, 2016; Granskog et al., 2017). Under present conditions of decreased ice concentration and thickness, an

influence of the ocean heat on the ice cover is increasing, providing positive feedback on a seasonal time scale (Ivanov et al., 2016). This effect is particularly important for the Atlantic sector of the Arctic Ocean, where inflowing warm waters facilitate an upward heat flux towards the ice base. Recent observations show an increased winter ventilation in the Nansen Basin and enhanced vertical heat flux, which reduced sea-ice formation in this region (Polyakov et al., 2017).

Observational data on ice concentration and extent are satisfactory since the advent of passive microwave satellite remote

sensing data in 1978 with a daily temporal resolution. Information on the evolution of ice thickness is, however, less accurate, consisting of submarine observations from several decades before 2000 and satellite remote sensing data during the last two decades. Passive and active microwave instruments provide information on multiyear ice coverage, which can be used as a proxy for ice thickness (Comiso, 2012). Since about 2004, more accurate information is available from satellite altimeters



applying lidars and radars at a resolution of about 25 km (Kwok et al., 2009). From the point of view of the atmospheric response to changes in sea ice cover, the most important sea ice variables are ice concentration and fraction of thin (less than 0.5 m) ice. Passive microwave L-band data from SMOS have shown unique capability to measure thickness of thin ice less than 0.5 m (Kalescke et al., 2012). Ice concentration is particularly important in conditions of a compact ice cover (> 90% ice

concentration) in winter (Lüpkes et al., 2008). Also the flaw polynyas along the Russian shelf in winter are important. They open and close repeatedly during the winter, depending on wind direction and speed, and causing new ice formation during opening and ice rafting and ridging during closing.

    Information on different ice types, floe size distribution, leads, and the snow pack on top of sea ice are collected during research cruises, ice stations and aircraft campaigns, as well as by satellite remote sensing methods. Considering exchange

processes at the air-snow, air-ice, snow-ice, and ice-water interfaces, such as surface and basal fluxes of momentum, heat, freshwater, $CO_2$, and $CH_4$, direct observations are very limited, mostly restricted to specific field campaigns with manned ice stations. However, data collected with sea ice mass-balance buoys allow possibilities for indirect estimation of the heat exchange at air-snow/ice and ice-water interfaces (Lei et al., 2018). The surface albedo is critical for the snow and sea ice mass balance during the melt season. It can be observed via remote sensing methods (Riihelä et al., 2013), but in-situ observations

are needed to develop better model parameterizations for the dependence of albedo on physical properties of snow, ice, and melt ponds (Perovich and Polashenski, 2012). Further, better observations are needed on light penetration through snow and ice, which is important for the ecosystems in and below the ice.

    Considering atmospheric and oceanic forcing on sea ice dynamics, the best source of information are simultaneous observations on the vectors of wind, ocean current, and sea ice drift (Leppäranta, 2011). In lieu of such data, sea ice drift

observations, based on buoys or satellite remote sensing, combined with estimates of the wind and ocean currents, based on atmospheric and ocean reanalyses, yield valuable information at least on regional scales (Spreen et al., 2011; Vihma et al., 2012). Small-scale processes of sea ice dynamics, including deformation, rafting, ridging, and breaking of ice flows, are more difficult to observe, but advance has been made using ice-station observations on the internal stress of the ice field (Weiss et al., 2007) as well as seismometer (Marsan et al., 2012; 2012) and ice radar observations (Karvonen, 2016). Radar observations

are good for detection of leads and ice ridges in areas where high-resolution SAR images (< 10 m resolution) are obtained. To cover larger areas, satellite remote sensing observations are needed, but challenges remain in detection of ice ridges. Storms have strong impact on sea ice deformation and create open leads and even break up during very cold condition (Itkin, et al., 2017).

    Large-scale evolution of the ice field results from a combination of thermodynamic and dynamic forcings. Quantification

of their relative contributions is still a challenge. This is mostly related to inaccuracy of sea ice thickness data. Also, the thermodynamic and dynamic forcings may often support each other, for example when strong winds advect warm, moist air masses to the over sea ice, simultaneously generating melt and ice advection away from the study region (Alexeev et al., 2017).



### 3.4 Ocean physics

Understanding the ocean heat and freshwater budgets is important for understanding the entire Arctic climate system, in particular its inter-annual and decadal variations. The Arctic Ocean and its marginal seas form a central element in the Arctic freshwater cycle, which is vital for the climate and ecosystems. Most physical, chemical and biological processes in the Arctic Ocean are influenced by the quantity and geochemical quality of freshwater (Carmack et al., 2016). Processes in the Arctic Ocean are closely linked with the atmospheric, terrestrial and extra-Arctic marine components of the climate system (Bhatt et al., 2014). However, the uncertainties in the heat and freshwater budgets of the Arctic Ocean and its marginal seas are not well quantified. Different studies have yielded different results (Carmack et al., 2016), but it is challenging to distinguish between differences originating from the lack and uncertainty of observations from those originating from temporal variations on inter-annual and decadal scales. In any case, there are evident lacks in the observation network.

The Arctic Ocean stratification is characterised by a stably-stratified low-salinity surface layer, which results from positive net precipitation and freshwater inflow from the Arctic rivers, Greenland ice sheet, and the Pacific through the Bering Strait (Rudels 2012). The thickness of the surface layer is limited by a strong halocline underneath and varies on seasonal-to-decadal time scales and across the basin. Freshwater is stored in the Arctic Ocean surface layer and either transported by the Transpolar Drift Stream out of the basin via the Fram Strait or accumulated in the Beaufort Gyre. Smaller amounts of freshwater are transported out of the Arctic via the Canadian archipelago. The sea ice transport is mainly wind driven, while the liquid freshwater transport occurs in buoyant boundary currents controlled by the planetary rotation.

Warm and saline Atlantic water flows into the Arctic Ocean mainly through Fram Strait in the West Spitsbergen current and St. Anna Trough. North-east of Fram Strait, the upper part of the Atlantic Water (AW) cools down due to heat loss to the atmosphere and freshens through mixing with melt water, thus transforming into less dense polar water mass. The undisturbed deeper portion of the AW does not noticeably change its properties, forming intermediate warm layer (150-900m) - the so-called Fram Strait AW (FSW). The other branch of the warm AW enters the Arctic Ocean by first passing over the Barents Sea, where it is extensively cooled and freshened by precipitation and mixing with melt water and riverine water. The main part of this Barents Sea branch water (BSW) enters the Arctic Ocean via the St. Anna Trough. North of the Laptev Sea on the Siberian shelf, the two Atlantic water branches meet and partly mix, but still keep their specific 'core' properties on further movement around the deep basin (Ivanov and Aksenov, 2013). In the Laptev Sea and further to the east low salinity shelf water originating from large Eurasian rivers admixes in the upper layer, strengthening the vertical density gradient in the cold halocline layer, which started to form after initial isolation of FSW from the ocean surface. The FSW partly recirculates towards Fram Strait along the western flank of the Lomonosov Ridge and partly enters Amerasian Basin making full cyclonic loop along the continental slope, and gradually cooling and freshening en route. Major part of the BSW also makes full loop around the deep basin (Rudels et al., 2014). Beneath the Atlantic Water is the Arctic deep water originating from cold, saline and dense Siberian shelf water plumes, the densest fraction of the BSW and the Norwegian Sea Deep Water inflow through Fram Strait (Jones et al., 1996). In a quasi-geostrophic balance, the deep water slowly flows in cyclonic boundary currents following



the topography of sub-basins (Nansen, Amundsen, Makarov and Canada basins) and returns towards Fram Strait. Flaw polynyas act as major producers of dense water, which may descend (cascade) to a substantial depth along and beyond the continental slope, ventilating various layers in the deep basin (Ivanov et al., 2004).

Tides and wind waves in the Arctic Ocean are important for the climate, coastal erosion, and navigation. Tides in the Arctic
Ocean have received relatively little attention compared to their potential importance. In particular the lunar semidiurnal component is strong in the Arctic shelf, with the amplitude reaching 2-3 m in the White Sea. On the basis of model experiments, Holloway and Proshutinsky (2007) suggested that tides can contribute to mixing of the Atlantic Water with overlying polar waters, further affecting the global ocean conveyer belt with potential impacts on the Arctic and global climate. The role of tides in mixing the Arctic Ocean has also been demonstrated by Luneva et al. (2015), who further showed that this has enhanced
the sea ice decline.

Decadal and inter-annual changes of wind wave fields in the Baltic, Barents and White seas in the period 1979-2010 have been estimated on the basis of the NCEP/CFSR reanalysis (Saha et al., 2010) and the numerical models SWAN (Booji et al., 1999) and WaveWatch III (Tolman, 2009). Information on the wave statistics and validation techniques applied are provided by Medvedeva et al. (2015), Myslenkov et al. (2015; 2017) and Korablina et al. (2016). The maximum of significant wave
height for the study period reached 8-9 m in the Baltic Sea, 15-16 m in the Barents Sea, and 4-5 m in the White Sea. Model experiments for storm surges in the Barents and White have shown that most of the highest surges were formed after the passage of Polar lows (Korablina et al., 2016). The Onega Bay in the White Sea and the Haipudyr Bay in the Barents Sea were found to be the areas of the most frequent formation of surges over the last decades.

In addition, there are several small-scale processes, which need to be parameterized in ocean and climate models. These
include exchange of momentum, heat, and salt at the ice–ocean interface, brine formation in ocean (Bourgain and Gascard, 2011), diapycnal mixing (Rainville et al., 2011), double diffusive convection (Sirevaag and Fer, 2012), as well as (sub-) mesoscale eddies and fronts (Timmermans et al., 2012). Quantification and understanding of these processes require high-quality process-level observations, but the temporal and geographical coverage of such observations is still very limited.

Multidisciplinary in situ data in the Arctic Ocean are still collected mainly during icebreaker expeditions, aircraft surveys,
and manned drifting platforms. However, these activities are irregular in time, very expensive, biased to the summer season, and hence poorly suited for providing regular long-term monitoring data. Moorings have been deployed at key locations in the gateways and rims of the Arctic Ocean (Figure 6), but they mainly deliver physical parameters from fixed depths in a delayed mode (Beszczynska-Möller et al., 2011). Nevertheless, NABOS measurements allowed documenting of Atlantic Water warm pulses in 2000s (Polyakov et al., 2011), and revealing strong seasonal cycle in the intermediate AW layer deep below the ocean
surface, which was not directly measured before (Ivanov et al., 2009; Dmitrenko et al., 2009). Biogeochemical and profiling sensors for moored applications are still very limited, resulting in insufficient multi-disciplinary data. Only in the Fram Strait, the key region for Arctic-Atlantic exchanges, a multi-disciplinary observatory (Hausgarten/FRAM) has been implemented for long-term ecosystem monitoring (Soltwedel et al., 2005). Extended spatial coverage of the upper Arctic Ocean is provided by



the Ice Tethered Buoys (ITP), which allow high resolution profiling in the upper 800-1000m and straightforward transmittance of data via satellite.

The availability of oceanographic data is comparatively good in the Barents Sea, Bering Sea, and Greenland/Norwegian Sea, whereas there are far less data from the less accessible central and eastern parts of the Arctic Ocean and Russian shelves. The SST field over the open ocean is fairly accurately known during the satellite era.

Process understanding and quantification of the state and changes of the Arctic Ocean circulation as well as heat and freshwater budgets are limited by the insufficient amount of observations. A specific challenge for in situ observations of the ocean is that only a part of the data are available in real time, whereas a lot of data can only be gathered when the instruments are recovered from the ocean.

### 3.5 Ocean chemistry and ecosystems

With increasing $CO_2$ partial pressure in the atmosphere, the capacity of the world oceans to uptake $CO_2$ continues to decrease as the reaction of $CO_2$ dissolution gradually tends to saturation. Under such conditions, the planetary greenhouse effect enhances. In turn, the ensuing surface ocean temperature growth leads to a shift of dissociated calcite, $CaCO_3$, to its solid phase (Chen and Tang 2012). Thus, the actual balance between dissociated and suspended phases of $CO_2$ becomes an issue of paramount importance (Seinfeld and Pandis, 2016). Shifts in the exchange of $CO_2$ between the aquatic medium and the boundary layer above are highly consequential also in terms of acidification of seas. In combination with the co-occurring external forcings, both processes are conducive to a variety of alterations in marine hydrobiological processes. Among the latter are the formation of nutrients uptakable by phytoplankton, rates of intracellular metabolism, primary production, and reshufflings in phytoplankton species composition and abundance (Bates and Mathis, 2009).

Figure 7 presents an example of a widespread phytoplankton bloom in the Barents Sea. Within the phytoplankton community, coccolithophores are the main producers of particulate inorganic carbon (PIC) in marine biosystems. Within this plankton group, Emiliania huxleyi is a particularly efficient producer of calcium carbonate in pelagic and coastal marine waters (Thierstein and Young 2004). It is obvious that driven by changes in the carbon cycle in the atmosphere-ocean system and ensuing climate change, the aforementioned alterations in the ecosystem of the world oceans are bound to affect the spatio-temporal dynamics of the coccolithophore growth (Schluter et al. 2014). Hence, investigation of the occurrence, spatial extent, and intensity of coccolithophore blooms across the world oceans is of multifaceted importance.

The available pioneer spaceborne studies (Kondrik et al., 2017) have revealed that blooms of Emiliania huxleyi in subpolar and polar marine environment occur annually and may cover hundreds of thousand square kilometres (e.g. up to $350 \times 10^3$ $km^2$ in the Barents and Bering Seas). Such blooms result in the release of huge amounts of particulate inorganic carbon (up to $7 \times 10^5$ kg in the aforementioned seas) and can last for 30-50 days, but in the Bering Sea the bloom duration is reported having lasted nearly throughout the year. Being not only photosynethysers, but also calcifiers, such algae are able to reduce the ability



of oceans to be efficient sinks of atmospheric $CO_2$ (Kondrik et al., 2018), thus, potentially, increasing the effect of global warming. Quantitative spaceborne investigations of this global phenomenon (as E. huxleyi blooms are omnipresent throughout the World Ocean) is mandatory for both accurate assessments of the actual carbon balance in the atmosphere–ocean system and projection of action of this mechanism over the decades to come through applying global and regional climatic models.

Nitrogen, phosphorus, iron and silicon are indispensable in primary production processes. Organic carbon is the principal forage for heterotrophic bacteria. Thus, the balance in input of the above substances controls the net carbon dioxide content in marine ecosystems. Allochtonous dissolved organic matter (ADOM) is also highly important in establishing the status quo of the light regime in such waters. The input and spread of the above elements are ultimately important for the marine ecosystem workings not only within the outfall of the major Eurasian rivers and adjacent shelf zones but across the entire Arctic Ocean.

Observations on the surface fluxes, carbonate system, other biogeochemical variables, and food chain are mostly restricted to scientific cruises and sparse coastal observations.

### 3.6 Linkages between the marine Arctic and Eurasian continent

The linkages between the marine Arctic and Eurasian continent can be broadly divided in three groups (a) large-scale atmospheric transports and teleconnections, (b) river discharge, and (c) other processes taking place within the coastal zone.

Considering (a), heat, moisture, pollutants, and other aerosols are continuously transported in the atmosphere between the Eurasian continent and the marine Arctic. Most of the transport is carried out by planetary waves and transient cylones, but also the mean meridional circulation, related to the Polar cell, contributes to the transports (Jakobson and Vihma, 2010).

Planetary waves include both propagating and quasi-persistent features in the atmospheric pressure field, such as the Siberian high-pressure pattern (Tubi and Dayan, 2013). Heat and moisture are transported both northwards and southwards, but the net transport across latitudes 60 and 70°N is northwards over most of Eurasia. However, southward net moisture transport occurs in summer in the belt between 40 and 140°E (Naakka et al., submitted). In addition to transports, planetary wave patterns generate teleconnections from the marine Arctic to Eurasian continent, as far as southern China (Uotila et al., 2014). Due to

the Arctic amplification of climate warming, individual cold-air outbreaks from the central Arctic to mid-latitudes have become less cold on the circumpolar scale (Screen, 2014). In winter, however, over large parts of Siberia the air is colder than over the ice-covered Arctic Ocean, and the latter serves as a source of relatively warm and moist air for Siberia. The sea ice loss from the Arctic Ocean has resulted in increased evaporation from the Arctic Ocean (Boisvert and Stroeve, 2015), and some studies suggest that this has caused increased snow fall in Siberia (Cohen et al., 2014).

Considering the coastal and archipelago zone of northern Eurasia, the atmospheric processes include coastal effects on the wind field, which are driven or steered by orographic and thermal effects (Moore, 2013). A remarkable change during recent decades is the intensification of the summertime frontal zone along the Siberian coast (Crawford and Serreze, 2016). In summer the terrestrial Arctic has warmed much faster than the marine Arctic (Figure 2), increasing the north-south temperature



gradient. However, the Arctic coastal frontal zone is not a region of cyclogenesis, but favours intensification of cyclones formed over Eurasia (Crawford and Serreze, 2016).

The present observation network is sufficient to detect synoptic-scale processes, but improvement is needed to detect coastal mesoscale flow features and to better quantify the magnitudes, vertical profiles, and trajectories of atmospheric transports.

Via river discharge, freshwater as well as dissolved and particulate matter are transported from the Eurasian continent to the Arctic Ocean. The freshwater discharge depends on the net precipitation (precipitation minus evapotranspiration) in the river basin. The total river discharge from Eurasia to the Arctic Ocean is almost 2000 km$^3$ per year, with an increasing trend since at least 1940s. Estimates of the six largest Eurasian rivers' inflow into the Arctic Ocean shows a 7% increase during 1936-1999 (Peterson et al. 2002) or +8,2 km$^3$/yr from 1948 to 2004 (Milliman and Farnsworth 2013). The Ob and Yenisey are the two largest rivers discharging into the Arctic Ocean (404 km$^3$/yr and 603 km$^3$/yr, respectively) and supplying the shelf seas with suspended and dissolved matter (AMAP, 1998). Both rivers account for about 40% of ADOM brought annually into the Arctic Ocean (Raymond et al., 2007). Degradation of permafrost lead recently to refractory organic carbon release, which has a significant impact into the present-day biochemical cycle and determines the role of river discharge into Arctic in both regional and global methane cycles (Shakhova et al. 2007). Riverine ADOM delivers, among others, organic carbon, inorganic nitrogen and phosphorus, iron, and silicates. Riverine particulate (seston) content in discharging waters is high (Hessen et al., 2010). However, not all studies on chemical riverine budgets include data from the spring high-flow period. Hence, existing datasets underestimate the fluxes of particulate heavy metals from the Siberian rivers to the Arctic Ocean (Chalov et al., 2014).

The drainage territories of the great Siberian rivers are in ongoing change, which includes several processes: thawing permafrost, ensuing growth of $CO_2$ liberation rates, as well as increased runoff, erosion and associated transport of total suspended mater and nutrients. Interaction of these processes in the changing climate system is complex, but we expect to see that increasing primary production and water turbidity will result in heat accumulation in the upper layers of the coastal ocean, strengthening of the thermal the stability, and shallowing of the thermocline. This will also cause some increase of alkalinity and buffering against $CO_2$ driven ocean acidification (Lenton and Watson, 2000).

Coastal erosion processes in the Arctic Ocean lead inter alia to inundation of the terrestrial coastal zone, which is due to wind driven breaking of the fast ice and exposure of the coast to marine wave action (Section 3.4), destruction of coastal forefront soil, and formation of a sloping bank. As a result, extensive areas of terrestrial permafrost become submarine permafrost. Because of ensuing warming, submarine permafrost starts thawing. The bottom thermal conditions thus changed, the processes of release of $CO_2$, methane, and other volatile substances from thawing submarine permafrost start developing on very large scales (Overduin et al., 2016). Despite the importance of this process, our present knowledge on submarine permafrost distribution, its thermal state, rates of greenhouse gas liberation and transport up into the atmosphere is meager (Ping et al., 2011). Also there is an evidence that terrestrial matter dominates in both water column and surface sediment of Arctic rivers compared to marine matter released from the sea floor (Karlsson et al. 2016). Bottom temperature, and water



salinity are acknowledged as important parameters controlling the above process. Closely related to the above factors are sedimentation and resuspension of coastal eroded matter, and its transport by wave action and currents.

## 4. Atmospheric and ocean reanalyses

The most complete information on the state of the marine Arctic climate system is based on combinations of observations and model results. Such combinations are produced via data assimilation to generate (a) analyses for initial conditions of operational forecasts and (b) reanalyses, where the same operational model version and data assimilation scheme is applied over a long historical period. Hence, reanalyses are more coherent in time, as the results are not affected by changes in the
operational model version and data assimilations method. Reanalyses consist of time series of the three-dimensional state of the atmosphere and ocean on a regular grid. Broadly applied atmospheric reanalyses include the global ones produced by European, U.S., and Japanese agencies and the regional Arctic System Reanalyses. A regional high-resolution reanalysis for the European Arctic is under work. Although these are the best sources of information on the past state of the Arctic atmosphere, reanalyses include also challenges, in particular in Polar regions, where the observational coverage is limited.
Major errors occur in radiative and turbulent surface fluxes (Tastula et al., 2013), near-surface air temperature and wind, as well as air moisture (Jakobson et al., 2012; Lindsay et al., 2014) and especially clouds (Makshtas et al., 2007; Lindsay et al., 2014). The problems are related, among others, in modelling of mixed-phase clouds and the ABL in stably-stratified conditions over ice and snow as well as in conditions of very heterogeneous surface temperature distribution.

Recently, ocean reanalyses have been constructed by many research groups. They combine observations with
hydrodynamical models to reconstruct historical changes in the ocean. Global and regional ocean reanalysis products are increasingly used in polar research, but their quality has only recently been systematically assessed (Uotila et al. 2017). First results reveal consistency with respect to sea ice concentration, which is primarily due to the constraints in surface temperature imposed by atmospheric forcing and ocean data assimilation. However, estimates of Arctic sea-ice volume suffer from large uncertainties, and the ensemble mean does not seem to be a robust estimate (Chevallier et al. 2017). On average, ocean
reanalyses tend to have a relatively low heat transport to the Arctic through Fram Strait and, as a result, cooler than the observed Atlantic water layer. These results emphasise the importance of atmospheric forcing, air-ocean coupling protocol and sea-ice data assimilation for the product performance.

The example illustrated in Figure 8 highlights the ocean reanalysis performance in terms of ocean salinity in the Eurasian basin. In the surface layer, the top 100 m, their salinities disagree the most due to differences in the surface layer freshwater
balance. The freshwater is originating from melted sea ice, atmospheric precipitation, river runoff and to a limited extent from the Pacific. Also, the amount of inflow of saline Atlantic water affects the basin salinity profile. Notably, the multi-product mean appears relatively close, although too fresh, to the observational products in contrast to many individual reanalyses. This feature is common to many climate model ensembles.



Large salinity disagreements in the surface layer do not co-vary with the corresponding temperature disagreements (not shown), which are the largest in the Atlantic Water layer below (300-700 m). The surface layer temperatures typically stay close to freezing around the year, and are also strongly constrained by the prescribed atmospheric near-surface temperatures used to drive many of the ocean reanalyses. For the products shown in Figure 8, these air temperatures are based on atmospheric reanalyses, mostly ERA-Interim. A notable exception is ECDA3 which is a coupled atmosphere-ocean product with the atmosphere relaxed towards NCEP-NCAR renanalysis. However, in addition to large ocean temperature discrepancies compared to other products (not shown), ECDA3 also has the largest salinity disagreement in the Eurasian Basin (Figure 8b). The accuracy of Eurasian Basin surface layer salinity in ocean reanalyses is strongly affected by the Siberian river runoff. Currently all reanalyses use a variety of adjusted runoff climatologies. This is clearly a shortcoming and could be one of the objectives that the PEEX researchers could aim to improve together. The use of interannually varying runoff data ideally based on all available observations would be a major step towards a more realistic Arctic Ocean reanalysis, in particular when combined with better precipitation, wind and temperature data from the latest atmospheric reanalyses.

## 5. Socio-economic evolution in the marine and coastal Arctic

The present socio-economic component of PEEX includes research on energy policy changes and their effect on the greenhouse gas emissions, especially in the Russian Arctic and Siberian regions (Lappalainen et al., 2016; 2018). In general, PEEX is interested in developing methods and concepts for integrating natural sciences and societal knowledge as a part of operational Earth System Sciences. At the moment we have a modelling framework, where we try to connect the energy consumption to emission models and current IPCC (RCP) scenarios and then run the climate models. Climate models provide input for the air quality, climate and aerosol predictions. This framework is relevant also for the marine and coastal Arctic. The marine Arctic is expected to become increasingly important from the socio-economic point of view, which will significantly broaden the socio-economic research activities of PEEX. The socio-economic importance of marine Arctic is related to perspectives for increasing navigation, fisheries, and oil and gas drilling.

Contemporary socio-economic conditions for the development of coastal areas of Arctic and northeastern Eurasia (coastal areas of the Bering and Okhotsk Seas) are characterized by considerable contrasts. The oil and gas areas of the Yamalo-Nenets and Nenets Autonomous Okrugs have a strong development momentum due to the development of new, non-depleted hydrocarbon fields and the implementation of new LNG projects addressing European and Asian markets. On the other hand, the coastal areas of Arctic Asia, including the Arctic regions of Yakutia, the territories of the Chukotka Autonomous Okrug, the Kamchatka Krai and the Magadan Region, are characterized by a significant outflow of the population for more than 25 years and the implementation of point-based and relatively short- and medium-term projects in gold mining as well as extraction of polymetals and coal.



The intermediate position between these poles of economic "success" and "depression" is occupied by the territories of the Murmansk Oblast, the Arkhangelsk Oblast and the northern parts of the Krasnoyarsk Territory, in which, for decades of industrial development, powerful territorial production complexes have been created in the sphere of maritime transport, mining industry (Murmansk Oblast), timber processing (Arkhangelsk Oblast), and mining (Norilsk industrial region, the

northern Krasnoyarsk Krai). All three of them have accumulated significant industrial material assets, and skilled human resources. At the same time there are many environmental problems inherent in these old industrial districts of the Russian Arctic. In the most accessible Murmansk and Arkhangelsk Oblasts, tourism has been developing for the last two decades, including international tourism. An important role belongs to mutual recognition of these territories under the umbrella of the Barents Region initiative, which is the example of the most successful and energetic cross-border cooperation in the

circumpolar Arctic.

Contrast is also characteristic for the situation in navigation issues along the Northern Sea Route. The last two years (2016 and 2017) have exceeded the peak of the Soviet-era transport in 1986, when 6.5 million tons were transported. However, if in the 1980s this was achieved due to a uniform operation along the entire Northern Sea Route, that is, through caravan wiring of ships "from Arkhangelsk to Anadyr" and further, now it is achieved mainly due to transportation work on the western parts

of the Northern Sea Route, where the development of new offshore and onshore hydrocarbon fields like Varandey, Prirazlomnoye, Novii Port, and the construction of a completely new port and the city of Sabetta with 30,000 inhabitants and the LNG plant have given a new momentum to the development of this coastal territory of the Russian Arctic. The situation is very different in the eastern sector (east of Dikson) of the Northern Sea Route. There are no major new projects onshore, although exploration work on the Kara Sea shelf, the Laptev Sea and, in the future, East Siberian and Chukchi Seas is and will

be steadily intensifying, primarily conducted by Rosneft. A completely different story is in the Sea of Okhotsk, where on the shelf of Sakhalin for more than 15 years there has been an industrial production of hydrocarbons for export markets to the nearest Asian countries.

Against the backdrop of the strongest polarization of the socio-economic development of the coastal areas of the Arctic and the northern Far East (northeast Asia), a common trend is emerging for all the territories – that is, "hydrocarbonization"

of economy. The economic profile of several territories, which were previously based on traditional reindeer husbandry and fisheries, is gradually beginning to shift to hydrocarbon economy under the influence of new discoveries of gas and oil both offshore and onshore. This will require very thorough and much more numerous distribution of stationary and mobile research activities of the natural environment and climate, their changes, and the impact of these changes on the risks of economic activity on land and at sea. Such integrated research has been conducted for many years in the delta of Lena, on the basis of

the Tiksi settlement. However, it seems that the scale of new economic development and the formation of absolutely new industrial regions on land and shelf of the Arctic, determined by the "Strategy for the development of the Arctic zone of the Russian Federation and ensuring national security for the period until 2020" (approved in 2017), will require much more intensive and regular research of the Eurasian Arctic facing the challenge of an entirely new wave of industrialization.



Examples of numerous and not completely understandable new environmental events like craters in the Yamal, unexpected releases of gas hydrates, frequent accidents of oil and gas pipelines under the influence of thawing permafrost demonstrate the need to increase interdisciplinary research efforts to understand the general patterns of development of natural-economic systems in the highly unstable modern Eurasian Arctic.

Another important and relatively new trend is the processes of gradual consolidation of the coastal municipal formations of Eurasian Arctic, as evidenced, for example, by the recent establishment of the Association of Coastal Municipalities of the Arctic Zone of Russia. Common challenges related, among others, to climate change and its effects on socio-economic stability of these territories will contribute to such consolidation.

Important research questions and challenges related to the socio-economic development of Eurasian Arctic under changing
climate include the following:

1. Is it possible to assert that the interaction of the ocean, land, atmosphere, and society in the coastal Eurasian Arctic is much more turbulent, unstable, and fluctuating than the interaction of natural and economic systems in the continental subarctic of Eurasia? To study this, analogous multidisciplinary research stations are needed in the Arctic seaside zone and in the
continental subarctic zone.

2. How and what is the difference in the ecosystem conditions provided by the natural landscapes of the land and ocean in the rapidly developing western part of the Northern Sea Route and in the stagnated eastern part of it? What are the regional differences in the models, intensity, and orientation of these interactions? For this, we need analogies and mobile expeditions simultaneously in the western and eastern parts of the marine and coastal Eurasian Arctic.

3. What exactly are the differences in natural-economic interaction in the development of oil and gas fields offshore and onshore? Again, stable and mobile research stations are needed that can monitor changes in gas and oil fields. Intuitively, it seems that the significantly greater specificity of gas as a natural asset, which causes the need to create a more complicated infrastructure complex immediately at the place of extraction and immediate transportation, carries higher risks and encumbrances on the natural environment than oil industrial complexes.

4. New large-scale projects for the creation of new Arctic ports (one of which has already been realized - the port of Sabetta) and transport corridors in the Eurasian Arctic also calls for a more detailed look at the entire dynamics of natural-economic interaction and the new instability that may arise in this regard.

5. In the Sakhalin Sea part of the Sea of Okhotsk there is experience of more than 15 years in the development of new industrial areas of the shelf. However, at present there is almost no research coordination between the activities in the Sakhalin region
and in the exploration of new oil and gas projects in the Arctic. Investigation of the similarities and differences of these regions would yield new knowledge on the Arctic specificity in the interaction of natural and economic systems at the land-sea margin.





6. With the general focus of MA-PEEX on the sea side, it is necessary to isolate the interaction of natural and economic systems on the narrow land-sea margin and broader issues of natural and economic interaction throughout the entire coastal zone. This is urgent, as many new offshore oil and gas projects are scheduled to be developed with the drilling base on the land, which is more economic and practical. Because of horizontal drilling, it is nowadays often more feasible to develop offshore deposit from the coast or from artificial islands (appropriate regulations are under work in Russia) than from a platform in the sea. Hence, research focus on the narrow land-sea margin is needed.

## 6. The way forward

It is vital that MA-PEEX will be developed with in close collaboration with SAON, AMAP, IABP, Arctic ROOS, Copernicus Marine Services, MOSAiC, and INTAROS. In addition to close international collaboration, the way forward includes opportunities arising from development of new technology, community-based observations, improved data management, and better atmosphere-ocean reanalyses.

### 6.1 Opportunities arising from new technology

Recent advances in observation technology generate improved possibilities to quantify the state of the atmosphere, cryosphere, and the ocean. Among others, there is potential for a more extensive application of Unmanned Aerial Vehicles (UAVs). Different UAVs are available for measurements ranging from the local ABL to circumpolar scales. Small, cost-effective UAVs can be applied to observed vertical profiles of air temperature as well as wind speed and direction up to 2-3 km (Reuder et al., 2012; Figure 9). Large sophisticated UAVs, such as the Global Hawk, can operate in circumpolar scales, also releasing dropsondes (Intrieri et al., 2014). The technology is developing fast and expected to continue so, but there are challenges related to financing of extensive UAV activities and to legal regulations, in particular for flights crossing the borders of national air spaces (AMAP, 2012). So far much less applied but highly potential observation device is Controlled Meteorological Balloon. The balloons allow regional observations as they can drift for a few thousand kilometres horizontally, and also take vertical soundings of wind and air temperature (Hole et al., 2016). The balloon performance is expected to increase with advance in micro-scale solar panels. Further, we expect better possibilities for atmospheric and Earth surface observations also via advance in performance and instrumentation of manned research aircraft. We also expect further advance in ground/ship/ice-based remote sensing of the Arctic atmosphere, as the methods introduced in Section 3.2 are progressively improving. Further, recent advance in satellite remote sensing has yielded better information on the temperature and humidity profiles over ice and snow (Perro et al., 2016).

There is promising development in autonomous ocean observing systems, which can significantly improve the capacity to collect data from the Arctic seas. Ice-Tethered Profilers (ITPs) provide high-quality upper-ocean observations available from the central Arctic throughout the year (Toole et al., 2011). ITPs offer a platform that can carry a cluster of instruments with



capability to transmit data via satellite in near real-time. Bio-optical sensor suites are developed for the ITPs for ecosystem monitoring (Laney et al., 2015). However, the ITP network is still sparse and covers only a limited ice-covered part of the Arctic Ocean. The Argo programme of oceanographic floats is the main observing system for the global ocean (Riser et al., 2016) but Argo floats relay on surface access, and are therefore not suitable for ice-covered Arctic regions. Only recently have the ice-capable RAFOS floats have been implemented (Klatt et al., 2007). Gliders have proven to be efficient for the upper-ocean measurements in many parts of the world but, as ARGO floats, they need open water for data offload and positioning. Gliders and floats operating in ice-covered regions have to rely on underwater geo-positioning systems (GPS). Regional acoustic networks for acoustic thermometry, underwater GPS, and passive acoustic observations (Mikhalevsky et al. 2015) have been used in Fram Strait (Sagen et al., 2016), and in the Beaufort Sea (Worcester et al., 2015). Gliders have been successfully operated under sea-ice in the Davis Strait (Lee et al., 2013) and were tested in Fram Strait (Sagen et al., 2017). However, the under-ice navigation of gliders is still in the development stage, and European gliders have not yet been tested in ice-covered Arctic seas.

Sea ice mass-balance buoys are already widely used to monitor the evolution of snow depth and ice thickness on ice floes drifting in the Arctic (Perovich et al., 2014). A new cost-effective type of mass-balance buoys consists of a high-resolution (2 cm) thermistor chain from the ocean through ice and snow to atmosphere (Jackson et al., 2103). An automatic algorithm has been developed to derive the snow depth and ice thickness from the temperature measurements (Liao et al., 2017). Advance is also expected via more extensive utilization of seismometer observations in sea ice research. These can record signals generated by ocean waves and swell propagating in sea ice, and yield information on the dependence of wave propagation on ice thickness (Marsan et al., 2012), which may further allow estimation of the average ice thickness and its evolution on a regional scale. Further, seismic measurements can complement satellite observations on sea ice deformation.

Observed shifts in river discharge and geochemical fluxes due to permafrost degradation which is not monitored in the existed scarce gauging network emphasize, importance of surrogate techniques in freshwater magnitude and quality observations. In particular, the remote sensing of both water runoff and water composition (SPM, CDOM and DOC) offers a powerful and reliable tool to enhance our understanding of hydrological impacts in major arctic river systems.

## 6.2. Opportunities arising from community-based observations

In all countries around the Arctic, there are community-based observing systems (Gofman 2010; Johnson et al. 2016; online atlas available at arcticcbm.org). With more people coming to the marine areas of the Arctic, there will be increasing opportunities for community members to contribute to better understand the marine Arctic ecosystems and their biotic and abiotic components (Eicken et al. 2014; Johnson et al. 2015; Nordic Council of Ministers 2015; Fidel et al. 2017; Johnson et al. 2018).




To understand the different potential uses and sources of community-based data on the marine Arctic, it is necessary to know the different kinds of community-based observing approaches that are used. These monitoring approaches range from programs involving community members only in data collection ("contributory citizen science" sensu Bonney et al. 2009) with the design, analysis and interpretation undertaken by professional researchers, to entirely autonomous monitoring schemes run by community members (Table 1; Danielsen et al. 2009).

Citizen science approaches where community members are involved only in data collection are particularly useful when large numbers of people are required to collect data across wide geographical areas and on a regular basis. This capitalizes on the strength of gathering the most data possible, even if the accuracy or precision of each individual data point may not be as high as that obtained by highly trained professionals. Monitoring approaches with more profound involvement of community members (the collaborative approaches in Table 1) are typically useful: (1) where community members have significant interests in natural resource use; (2) when the information generated can have an impact on how one can manage the resources and the monitoring can be integrated within the existing management regimes; and (3) when there are policies in place that enable decentralized decision-making.

To illustrate the potential uses of data from community based observing in marine areas of the Arctic, we provide below an example from Greenland. The Greenland Government has piloted the development of a simple, field-based system for observing and managing resources developed specifically to enable Greenlandic fishers and hunters to document trends in living resources and to propose management decisions themselves (Danielsen et al. 2014; searchable database available at https://eloka-arctic.org/pisuna-net/). The system was designed to build upon existing informal observing methods, and it includes most of the aspects that are believed to make knowledge generation initiatives 'culturally appropriate' (Pulsifer et al. 2011). At the national level in Greenland, there is considerable scope for collecting community member observations from this system and using them to track wider trends in the abundance of resources while at the same time increasing community members' voice in higher-level decision-making (Table 2). Data from community-based observing could potentially be aggregated to generate larger-scale overviews of, for instance, species range and phenology, habitat condition, opportunities and threats, the impacts of management interventions, and the delivery of benefits such as wildlife resources to the community members from the natural ecosystems.

As well as providing data to inform natural resource management decisions, community-based observing has the potential to shed valuable light on environmental changes at national and even pan-Arctic scales (Huntington et al. 2013; Chandler et al. 2016). The Greenland example described above is one such system currently in development, which has been explicitly designed to allow such upwards movement of data, and ultimately to permit larger-scale analyses. To the extent that systems like this can be implemented and replicated, important gaps in the monitoring of coastal areas of the Arctic seas can be plugged, at relatively low cost, while at the same time increasing community members' input to higher-level decision-making.

Most importantly, for community-based information to be useful at larger scales, monitoring schemes will need to be established in more sites and regions (Danielsen et al. 2005). Results can also only be synthesized where many programmes



have monitored the same attributes. They need not all use a single standardized technique – this would be difficult given the importance of the monitoring schemes being autonomous, and would preclude schemes from being responsive to local circumstances and needs. However, it is important that only a relatively small number of methods, each well replicated, is used across the set of studies to be analysed. Provided this is the case then meta-analytical techniques can be used to check (and if 5 necessary adjust) for differences in results being due to differences in field methods.

**6.3 Opportunities arising from improved data management**

There is increasing international collaboration in Arctic marine data collection and data management. Initiatives such as PEEX, 10 AMAP, IABP, Arctic ROOS, Copernicus Marine Services, INTAROS, as well as the SAON committees ADC (Arctic Data Committee) and CON (Committee on Observations and Networks) will all contribute to the overall collection of data as well as dissemination and management of data from the Arctic. MA-PEEX is expected to particularly benefit from the INTAROS work to further develop coordination and collaboration between data providers in the pan-Arctic region in order to better use existing systems and resources from many countries.

15 The Arctic in situ data are managed in a large diversity of levels, reflecting the many types of observing systems, which differ in the technical solutions adopted and in the maturity and organization of their various components. For instance, in the atmospheric domain there are several mature observing systems, such as international networks, that follow standardized data managements. In the marine sphere observations are more diversified and fragmented, providing more types of data with various degree of standardization. The marine observing systems are usually identified on the basis of the platforms utilized, 20 such as moorings, floats, and gliders (GCOS, 2016). Advance in data management can be made by building connections between distributed data repositories.

After the Arctic Science Summit Week in Helsinki, Finland, in April 2014, the Arctic Data Committee (ADC) was created by merging the Data Standing Committee (IDSC) of the International Arctic Science Committee (IASC) and the Committee on Data and Information Services SAON. The purpose ADC is to promote and facilitate international collaboration towards 25 the objective of free, ethically open, sustained and timely access to Arctic data. To reach this objective, ADC will support the international community to adopt, implement and develop (where necessary) data and metadata standards. ADC lists several existing multidisciplinary databases of Arctic data and metadata, such as the Polar Data Catalogue (PDC).
The Arctic Portal is a gateway to Arctic information, providing visualization of Arctic data and links to the main Arctic data repositories. These are the National Science Foundation Arctic Data Center (NSF-ADC), the National Snow and Ice Data 30 Center (NSIDC), the Global Cryosphere Watch (GCW), the ACD Arctic Coastal Dynamics, the Arctic Hydrological Cycle Monitoring, Modelling, and Assessment Program (Arctic-HYDRA), the IASC Network on Arctic Glaciology. Each of them provides integration, standardization, and access to Arctic data in their specific thematic area. The Arctic Portal has also





developed a Data Management System (DMS) for permafrost monitoring parameters of the Global Terrestrial Network for Permafrost (GTN-P).

Major advance has also taken place in the data management of the Svalbard Integrated Arctic Earth Observing System (SIOS). The data centers presently accessible through SIOS are: the Arctic Data Archive System (Japan), the Arctic Data Center (Norwegian Meteorological Institute), PANGAEA (Alfred Wegener Institute, Germany), the British Antarctic Survey (UK), the Italian Arctic Data Center (Italy), the EBAS database (Norwegian Institute for Air Research (NILU), Norway), the Norwegian Marine Data Centre (Norway), the Norwegian Polar Data Center (Norwegian Polar Institute, Norway), the Norwegian Satellite Earth Observation Database for Marine and Polar Research (NORMAP). Other multidisciplinary web data portals with European focus but covering also the Arctic or part of it provide single access point to thematic data archived in various local, national, regional and international repositories. These portals are not just a link to the existing data, but also provide quality-assurance and standardization of the observations, as well as processed data products. The data accessible through these gateways are interoperable and free of restrictions on use. They are the European Marine Observation and Data Network (EMODnet), the ACTRIS Data Centre web portal, and the ICOS Carbon Portal.

The Global Atmosphere Watch (GAW) Programme maintains and applies long-term systematic observations of the chemical composition and related physical characteristics of the atmosphere, emphasizing quality assurance and quality control, and delivering integrated products and services related to atmospheric composition. The GAW World Data Centers collect, document and archive atmospheric measurements and the associated metadata from measurement stations world-wide and make these data freely available to the scientific community. In some cases, WDCs also provide additional products including data analyses, maps of data distributions, and data summaries.

The MA-PEEX community will benefit from all these advances in data management, and will itself adopt the best management practices. To ensure that research data are soundly managed, the European Commission has recently published data management guidelines for the Horizon 2020 projects (EU, 2016). The guidelines help to make the research data findable, accessible, interoperable and reusable (FAIR). The application of the FAIR data principles (Wilkinson et al., 2016) requires that the data are accompanied by rich metadata and are uniquely identified by persistent identifiers. Although the FAIR principles are not yet implemented in most of the existing Arctic marine data, they will be applied as much as possible for the multidisciplinary data produced in MA-PEEX.

## 6.4. Opportunities arising from better reanalyses

With more powerful computational resources, the numerical atmosphere-ocean models can be run with higher precision being able to resolve smaller flow features with less need for a sub-grid-scale parameterisation. For example, the ocean models have relatively recently moved on to use eddy-resolving configurations, which for mid- and high latitudes, and shallow regions require better spatial resolution than 10 km (Hallberg, 2013). Dynamically, these ocean eddies have the corresponding





importance for the ocean flow than cyclones for the atmospheric flow. Therefore, significant improvements in the realism of ocean simulation are expected as the ocean models increasingly and routinely start resolving these scales.

More sophisticated data assimilation methods are constantly being developed and applied in the latest reanalysis products. For example, reanalysis systems are increasingly moving from the 3DVAR data assimilation to 4DVAR (adjoint) assimilation,

and towards the ensemble forecasting. Here ensembles consist of multi-product ones (e.g. Vitart et al. 2017). In particular, the sea-ice data assimilation methods have been developed fast lately, with the latest approaches utilising adjoint methods and sea-ice thickness observations, not only sea-ice concentration (e.g Koldunov et al. 2017). Due to the long memory of sea-ice thickness, a significant improvement of forecast skill extending to inter-annual scales can be expected.

Finally, coupled reanalyses products are becoming increasingly available. They realistically resolve air-ice-ocean

interactions compared to their stand-alone atmosphere and ocean counterparts. First comparisons show that the coupled products stand out in particular in terms of surface fluxes and near-surface variables (Figure 8, Zhang et al. 2017, Uotila et al. 2017), but as has been the case with regards to coupled climate modelling, one can expect that their realism will quickly improve due to intensive development efforts. Clearly the coupled atmosphere-ocean approach is the way of the future in the construction of reanalysis and forecasts.

The emergence of large number of atmosphere, ocean and coupled reanalysis products shows a major promise and they are becoming an increasingly valuable resource for researchers of the marine Arctic. However, atmospheric reanalysis include errors and inaccuracies in the Arctic, in particular in the ABL. Better quality and more dense observation network should help in resolving this issue, but there is also a need to improve the physical parameterizations of the models (Vihma et al., 2014). Significant improvements in ocean and coupled reanalysis products are upcoming, mainly due to their relative immaturity at

the moment when, compared to the atmospheric ones. These improvements will be a result of more multivariate and advanced data assimilation approaches and increasing model resolution.

## 7. Discussion

The knowledge on physical and biogeochemical processes in the Arctic Ocean is limited, partly because the sea ice cover severely hampers observations, both in the upper layers and deep waters. There is a severe lack of in situ multidisciplinary, in particular biogeochemical, data for the Arctic Ocean, and significant patterns of the Arctic ecosystem are currently not well monitored. To improve the ocean component of an Arctic observing system is a pronounced technological and logistical challenge. In the design of MA-PEEX, the SMEAR concept, successfully applied in PEEX (Lappalainen et al., 2018), can be

applied in coastal and archipelago stations, such as Tiksi, Cape Baranova, Ny Ålesund, Barentsburg, and Villum Station Nord. In addition to research on coastal processes, observations taken at such regular stations provide a valuable but perhaps still underutilized opportunity to study the atmosphere over the Arctic Ocean. Systematic analyses of data gathered close to the



shoreline in conditions of onshore winds may reveal a lot of new information on variations and processes in the marine atmosphere on time scales ranging from daily to climatological.

A key question in the design of MA-PEEX is, if the SMEAR concept is applicable for the Arctic Ocean itself. It will probably be more cost-effective to further develop a strongly distributed marine observation network, based on enhancement

of programs such as IABP, ARGO, and Arctic ROOS. The trend in marine observations, both globally and in the Arctic, has been towards increasing application of autonomous buoys, moorings, AUVs, and UAVs, and the relative importance of centralized observations in research vessels and ice stations has simultaneously decreased. In the Arctic these trends have been enhanced by the sea ice decline. For example, the pivotal Russian ice station program has not been continued since 2015. The MOSAiC expedition from autumn 2019 to autumn 2020 will be based on a major ice station, supported by R/V Polarstern,

and can be regarded as comparable to terrestrial SMEAR Flagship stations, although with a duration of a year only. After MOSAiC we will better understand, if there are possibilities for continuation of such activities in the Arctic Ocean, where the ice extent and thickness will continue to have a decreasing trend. One of the future possibilities could be an ice-resistant platform, planned to be built in Russia.

Particularly important is that MA-PEEX will be well integrated with the existing atmospheric, terrestrial, and socio-

economic components of PEEX. This requires special attention to the linkage processes, such as atmospheric teleconnections and transports in and out of the Arctic, river discharge and related transports of dissolved and particulate matter, as well as various coastal processes. There is a need for integration of long-term monitoring, modelling, and process studies to better understand and quantify the atmospheric and riverine transports (Prowse et al., 2015). Better integration of observations and models is needed to understand the entire fluvial system's response to hydro-climatological changes, particularly related to

changes in the biochemical cycle due to permafrost thaw.

In addition to this integration, specific attention is needed to improve and enhance the following activities.

● International collaboration with other programmes and projects, such as AMAP, SAON and INTAROS, is vital for MA-PEEX in further development of observation networks, new observation methods, community-based observations, data management, as well as models and methods for climate research, operational services, and atmosphere-ocean reanalyses.

The opportunities for the advance of MA-PEEX will arise from international collaboration in these fields.

● For community-based information to be useful for MA-PEEX at larger scales, monitoring schemes should be established in more sites and regions in the Eurasian Arctic. Some A number of community-based activities have already been established in the Arctic regions of Russia, Norway and Finland (Johnson et al., 2016; Danielsen et al. 2017), but the examples from Greenland, presented in Section 6.3, may serve as a guideline for further advance.

● Continuous effort is needed to develop better reanalyses, as they provide the best available information on the state and past changes of the marine Arctic climate system. Improvement of reanalyses requires better spatial and temporal coverage and higher accuracy of observations, as well as better models and data assimilation techniques. However, there are numerous



variables, above all related to atmospheric composition and ocean biogeochemistry, which are not included in presently available reanalyses. Advance in observations is even more crucial for our knowledge on those variables.

● Better weather and marine services are needed to enable environmentally and socially responsible growth. The environmental risks associated with in transport flows across the Arctic are closely tied to adequate anticipation of adverse weather and ice conditions. How and to what extent the Arctic service level will unfold depends also on the international cooperation regarding regulations and their enforcement regarding environmental protection and transport safety in the Arctic. Better cooperation reduces uncertainties for more ambitious development of weather and marine services and thereby for the development of transportation flows across the Arctic.

● Socio-economic evolution in the marine, coastal, and terrestrial Eurasian Arctic includes a lot of uncertainties and distinct regional differences. We can identify different socio-economic perspectives, challenges, and opportunities between the Russian, Scandinavian, and Greenlandic parts of the MA-PEEX region, between the marine and continental Russian Arctic, as well as between the Sakhalin area and the western and eastern parts of the coastal zone of the Arctic Ocean. Despite of the differences, in all these regions there is a strong need for cross-disciplinary research to obtain comprehensive understanding on the interactions between the physical climate system, ecosystems, and socio-economics, which all are changing rapidly.

● We need well-coordinated data management of Arctic multidisciplinary in situ observations. The implementation of the FAIR principles in data management is envisaged, which requires permanent data archiving and easy open access.

MA-PEEX will promote international collaboration, sustainable marine meteorological, sea ice, and oceanographic observations, advanced data management, and multidisciplinary research on the marine Arctic and its interaction with the Eurasian continent.

*Author contributions.* T. Vihma led the design and coordination of the manuscript and wrote a major part of it. P. Uotila wrote Sections 4 and 6.4, and prepared Figure 8. S. Sandven wrote large parts of Section 3.4 and contributed to several other sections. D. Pozdnyakov wrote most of Sections 2.2 and 3.5. A. Makshtas, V. Ivanov and I. Frolov contributed to Section 3 and planned Figures 4 and 5. A. Pelyasov wrote Section 5. R. Pirazzini contributed to Sections 3.1 and 6.3. F. Danielsen and A. Albin wrote Section 6.2 and prepared Tables 1 and 2. H. Lappalainen contributed to Sections 2.2 and 3.5. S. Chalov, S. Dobrolyubov, V. Arkhipkin, and S. Myslenkov wrote parts of Sections 3.1 and 3.6. B. Cheng contributed to Section 3.3. The idea to write the manuscript came from M. Kulmala. He, together with H. Lappalainen and T. Petäjä, contributed to integration of the plans of MA-PEEX to the activities of the other components of PEEX.

*Competing interests.* The authors declare that they have no conflict of interest.





*Acknowledgments.* We thank O. V Mugdaba and A. L. Garmanov from the Arctic and Antarctic Research Institute for preparing Figures 4 and 5, respectively. We express our gratitude for the financial support of this study provided by the Academy of Finland (contract 283101, Vihma and Cheng), the EC H2020 project INTAROS (grant 727890, Sandven, Pirazzini, Danielsen, Albin), the EC Marie Curie Support Action LAWINE (grant 707262, Uotila), the Russian Science

Foundation (RSF, projects 14-27-00083, 14-37-00038, 14-37-00053-П, and 17-17-01117, Pozdnyakov), the Russian Fund for Basic Research (project 17-29-05027), and the Ministry of Education and Science of the Russian Federation (Project RFMEFI61617X0076, Makshtas). With respect to Figure 7, we acknowledge the use of Rapid Response imagery from the Land, Atmosphere Near real-time Capability for EOS (LANCE) system operated by the NASA/GSFC/Earth Science Data and Information System (ESDIS) with funding provided by NASA/HQ.

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

Table 1. Arctic and sub-Arctic natural resource monitoring schemes across a spectrum of possible monitoring approaches based on the relative participation of different actors (modified from Danielsen et al. 2009; Huntington et al. 2013). The relative role of community members in the monitoring systems increases from bottom to top between the five categories of monitoring systems.





| Category | Arctic examples | Description | | |
|---|---|---|---|---|
| Fully autonomous local monitoring | Customary conservation regimes, e.g., in Canada (Ferguson et al. 1998, Moller et al. 2004) | The whole monitoring process – from design, to data collection, to analysis, and finally to use of data for management decisions – is carried out autonomously by local stakeholders | | |
| Collaborative monitoring with local data interpretation | Arctic Borderlands Ecological Knowledge Co-op, Canada (Eamer 2004); Community-based monitoring by Inuvialuit Settlement Region, Canada (Huntington 2011); Opening Doors to the Native Knowledge of the Nenets, Russia (www.arcticcbm.org); Piniakkanik Sumiiffinni Nalunaarsuineq (PISUNA), Greenland (Danielsen et al. 2014; www.pisuna.org) | Locally based monitoring involving local stakeholders in data collection, interpretation or analysis, and management decision making, although external scientists may provide advice and training. The original data collected by local people remain in the area being monitored, but copies of the data may be sent to professional researchers for in-depth or larger-scale analysis | | |
| Collaborative monitoring with external data interpretation | Integrated Ecosystem Management (ECORA), Russia (Larsen et al. 2011) | Local stakeholders involved in data collection and monitoring-based management decision making, but the design of the scheme and the data analysis and interpretation are undertaken by external scientists | | |
| Externally driven monito-toring with local data collectors | Environmental Observations of Seal Hunters, Finland (Gofman 2010); Fávllis Network, Norway (Gofman 2010); Monitoring of breeding eider *Somateria mollissima*, Greenland (Merkel 2010); The Piniarneq fisheries catch and hunting report database, Greenland | Local stakeholders involved only in data collection stage, with design, analysis and interpretation of monitoring results for decision-making being undertaken by professional researchers, generally far from the site | | Increasing role of community members |
| Externally driven, researcher executed monitoring | Multiple scientist-executed natural resource monitoring schemes with no involvement of the local stakeholders | Design and implementation conducted entirely by professional scientists who are funded by external agencies and generally reside elsewhere | | |



**Table 2.** Comparison of community members' perceptions and trained scientists' assessments of trends in the abundance of 18 marine attributes in NW Greenland 2009-2011 (Danielsen et al. 2014). Legend: ↑, increased abundance; ↓, declining abundance; ⇔, no major change in the abundance; ‡, increased abundance reported in some areas, decline in other areas; Few data, there are little or no abundance data available; ✓, correspondence between community members' and scientists' assessments; (✓), probable correspondence between community members' and scientists' assessments but the time, area and/or temporal/spatial scale of the assessments do not match; ⊘, no correspondence. D, Disko Bugt; N.a., not applicable; U, Uummannaq Fjord. *For latin names and details see Danielsen et al. (2014) and https://eloka-arctic.org/pisuna-net/.

| | Attributes | Percep-tions* | Scientists' assessments | Source of scientists' assessments* | Correspon-dence |
|---|---|---|---|---|---|
| Fish | Atlantic cod, D | ‡ | Few data | Siegstad 2011 | N.a. |
| | Wolffish *spp.*, D | ↑ | ↑/⇔ | Siegstad 2012 | ( ) |
| | Greenland halibut | ↑ | ↓/⇔ | Siegstad 2011; 2012 | ⊘ |
| Marine mammals | Ringed seal | ↓ | Few data | Boertmann 2007; Rosing-Asvid 2010 | N.a. |
| | Harp seal, D | ↑ | ↑ | Department of Fisheries and Oceans 2010; Rosing-Asvid 2010 | |
| | Narwhale | ‡ | Few data | North Atlantic Marine Mammal Commission 2012 | N.a. |
| | Humpback whale | ↑ | ↑ | Heide-Jørgensen *et al.* 2011 | ( ) |
| | Minke whale, D | ↑ | ↑ | Heide-Jørgensen *et al.* 2010 | ( ) |
| | Minke whale, U | ⇔ | Few data | No information | N.a. |
| Birds | Common eider | ↑ | ↑ | Chaulk *et al.* 2005; Merkel 2010 | ( ) |
| | White-tailed eagle, D | ↑ | Few data | No information | N.a. |
| | Large gulls*, D | ↑ | Few data | Boertmann 2007 | N.a. |
| | Arctic tern, D | ↑ | ⇔ | Boertmann 2007; Egevang & Frederiksen 2011 | ⊘ |
| | Brünnich's guillemot, breeding | ↓ | ↓ | Burnham *et al.* 2005; Labansen & Merkel 2012 | |
| | Little auk, D | ↑ | Few data | Egevang & Boertmann 2001; Boertmann 2007 | N.a. |
| Other | Winter sea-ice*, U | ↓ | ↓ | Danish Meteorological Institute | |
| | Offshore ships, U | ↑ | ↑ | Arctic Marine Shipping Assessment 2009 | ( ) |
| | Trawling, D | ↑ | Few data | No information | N.a. |



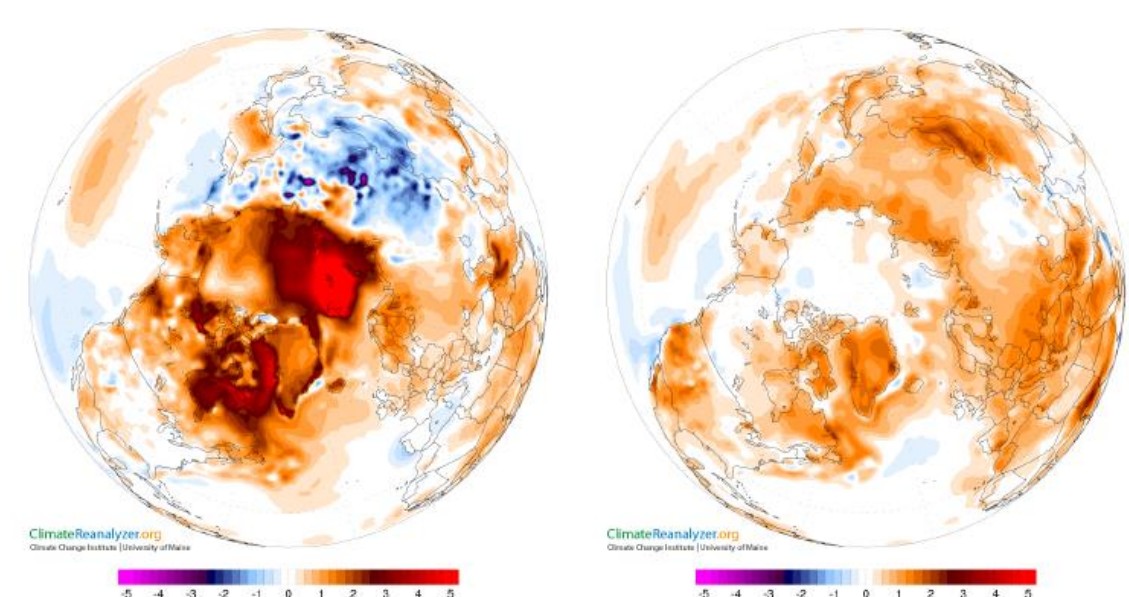

**Figure 1.** Schematic illustration of the marine Arctic component of PEEX (MA-PEEX)

**Figure 2.** Differences in winter (DJD, left panel) and summer (JJA, right panel) 2-m air temperature between the periods 2000-2015 and 1979-1999 according to ERA-Interim reanalysis. Figure drawn applying Climate Reanalyzer.

**Figure 3.** Distribution of buoys belonging to the International Arctic Buoy Programme in May 2018. The codes refer to the variables measured by different buoy types. Reproduced from http://iabp.apl.washington.edu/ monthly_maps.html.





**Figure 4.** The Russian Arctic Zone and hydrological network of Roshydromet in 2017. The codes are as follows: 1: the Russian Arctic Zone according to the Presidential decree №296 from 2 May 2014 and №287 from 27 June 2017; 2: the southern administrative boarder of the Arctic zone; 3: active marine hydro-meteorological shore network (3a – conserved stations); 4: active hydro-meteorological network of surface water objects on land (4a – conserved stations), and 5: locations of AARI hydrological studies.





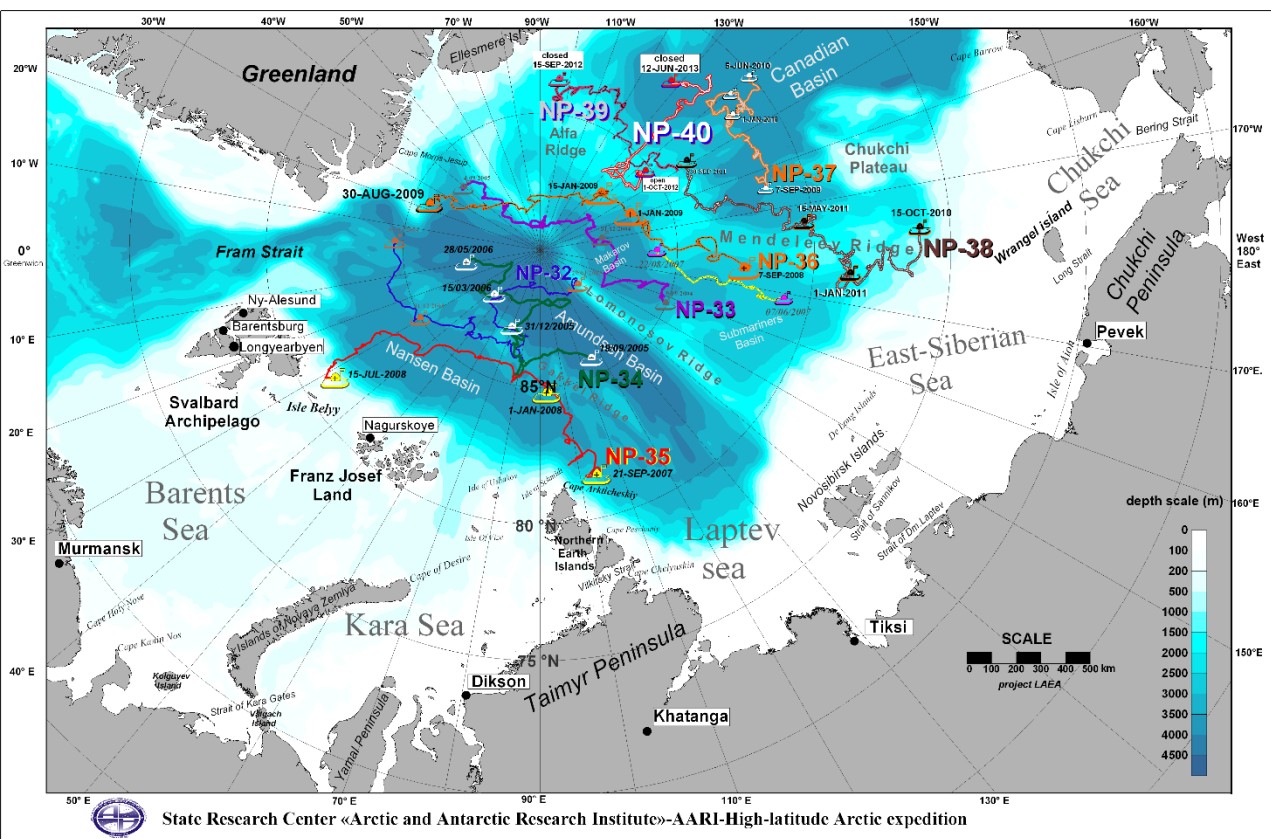

**Figure 5.** Trajectories of Russian "North Pole" drifting stations in 21st century.



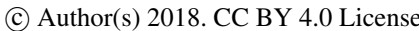

**Figure 6.** Oceanographic observations carried out during the Nansen and Amundsen Basins Observational System (NABOS) cruise in summer 2015, including CTD profiles, biological stations, deployment and recovery of moorings, as well as deployment of buoys and gliders. Source: http://research.iarc.uaf.edu/NABOS2/cruise/2015/. Reproduced with permission from Igor Polyakov, the University of Alaska Fairbanks.

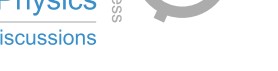

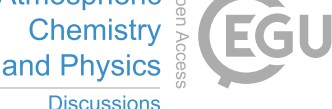



**Figure 7.** A phytoplankton bloom in the Barents Sea acquired by the Moderate Resolution Imaging Spectroradiometer (MODIS) on the Terra satellite on 6 July 2016. The phytoplankton may contain coccolithophores. The image is from the Rapid Response imagery from the Land, Atmosphere Near real-time Capability for EOS (LANCE) system operated by the NASA/GSFC/Earth Science Data and Information System (ESDIS).



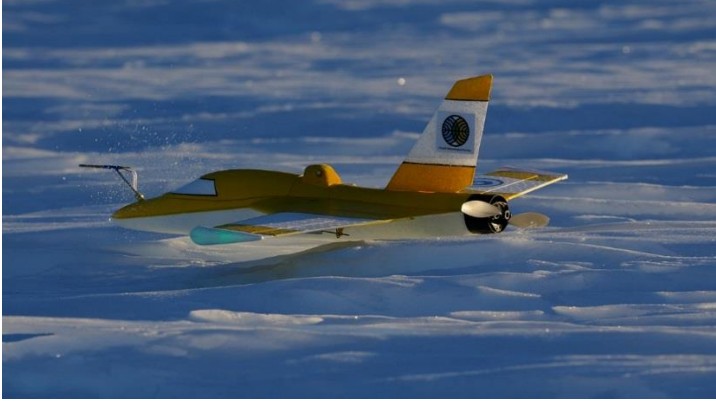

**Figure 8.** (a) Average surface salinity in based on Sumata et al. (2017) observed climatology, (b) mean departure of four ocean reanalysis from the climatology selected from Uotila et al. (2017), and (c) the salinity spread of four ocean reanalysis. Figure illustrates that the Arctic Ocean salinity uncertainty is the highest on the Siberian shelf, in particular close to the large rivers. This high uncertainty highlights the need for more measurements from the region.

**Figure 9.** Small Unmanned Meteorological Observer (SUMO), which is used to measure vertical profiles of air temperature, humidity and wind speed up to the height of 2-3 km. Photo Priit Tisler, Finnish Meteorological Institute.