# Peer review of "Towards an advanced observation system for the marine Arctic in the framework of the Pan-Eurasian Experiment (PEEX)"

_Atmospheric Chemistry and Physics, 2018_

## Referee Comment (RC1) · Anonymous Referee #1 · 24 Jul 2018

The paper introduces the need to study in depth the marine part of the Eurasian arctic, supplementing the ongoing effort of PEEX, that addresses the terrestrial part. The document is a compilation of recent work (the past two decades) on the area, a list of relevant scientific subjects, including the corresponding observational component, and a final part of different proposition to push the research in the area forward, including novel ways of data production and societal impacts. In general the paper is interesting, but it is too long, the different chapters have a lot of superposition and the list of references is incomplete (cites in the text not in the list) or with errors in the references. The impression is that this is a report of the interested group that has been submitted as a paper to ACP. The authors should reduce the text indicating the main points, including clearly the rationale and the actions to perform. This is why I propose a return of the

paper to the authors encouring resubmission of a much shorter document. I think it is easier for the authors than to request Major Revisions providing a very long list of Major Comments. Below I summarize some points according to the ACP review guidelines.

1. Does the paper address relevant scientific questions within the scope of ACP? The subject is correct in introducing the need to perform the study in the marine Eurasian area, but it fails to focus on what is really needed. It does not provide a clear rationale on how to proceed neither what actions are essentially needed.

2. Does the paper present novel concepts, ideas, tools, or data? The paper tries to identify all novel methods of measuring to see if they would be of use for their effort. Nevertheless they do not proceed to analyze how these methods should be implemented in such harsh environment.

3. Are substantial conclusions reached? No, other than the experiment is necessary (and I agree with that).

4. Are the scientific methods and assumptions valid and clearly outlined? The methods are well described, the assumptions are very reasonable. However, the second half of the paper dealing with the societal impact is not clear. Sections 5 (Socio-economy) and 6 (The way forward) are long, too wordy and should be drastically reduced or be written in a more clear manner.

5. Are the results sufficient to support the interpretations and conclusions? There are not results in the paper, but there is a very good compilation of the work done, so the authors are in very good position to go for the experiment, but this is more for a research proposal than for a scientific paper.

6. Is the description of experiments and calculations sufficiently complete and precise to allow their reproduction by fellow scientists (traceability of results)? Not applicable, since it is a review/prospective paper.

7. Do the authors give proper credit to related work and clearly indicate their own

new/original contribution? Yes, the compilation of previous work is excellent, although the reference list misses a lot of the references in the text and some references are inconsistent between the text and the list.

8, Does the title clearly reflect the contents of the paper? Yes.

9. Does the abstract provide a concise and complete summary? The abstract is good.

10. Is the overall presentation well structured and clear? The structure of the paper is good, but every section is too long and there are too many differences between sections, some are reltively concise and very clear others are more speculative and less precise. It is true that the paper addresses many different subjects and it is a big challenge to make a readable document with such a varied contents. However I am positive that a second try will manage to produce a good paper on MA-PEEX.

11. Is the language fluent and precise? In general English is good although there are some language mistakes, easy to correct. There are some changes of writting style between sections, probably due to different contributors, but the main problem is that not every section is written with the same approach, some are good others are weak.

12. Are mathematical formulae, symbols, abbreviations, and units correctly defined and used? Not applicable, since no mathematical expressions or formulae are used in the text or in the figures.

13. Should any parts of the paper (text, formulae, figures, tables) be clarified, reduced, combined, or eliminated? As indicated, the paper should be substantially reduced in size and also be written in a more internal coherent manner.

14. Are the number and quality of references appropriate? See point 7 on the errors in the references. The list is extensive but appropiate, it reflects well the scientific work made in the area in the recent times.

15. Is the amount and quality of supplementary material appropriate? No supplementary material that I can see. However if the authors produced a shorter version,

perhaps they could derive some of the removed parts to supplementary material.

---

## Referee Comment (RC2) · Anonymous Referee #2 · 23 Aug 2018

The manuscript by Vihma et al. describes a marine Arctic component (MA-PEEX) as supplement to the existing terrestrial/atmospheric the Pan-Eurasian Experiment (PEEX) and coastal the SMEAR (Station Measuring Ecosystem-Atmosphere Relations) concepts. The authors' main selling point is that they are going to investigate the behavior of the physical and biogeochemical processes to improve the ocean component of an Arctic observing system based on the marine meteorological, sea ice, and oceanographic observations for the Arctic Ocean itself. In the same time, the SMEAR concept can be applied in coastal and archipelago stations, such as Tiksi, Cape Baranova, etc. The topic of the study is highly interesting and has potential. I also think that the exercise has an obvious practical value. However, I have concerns about the manuscript.

[Figure]

In my view, this document is more like a proposal or 'letter of intent' rather than usual research paper. Ultimately, this is an editorial decision, but the topic appears more appropriate for another journal than ACP. For example, I used to see such papers in Bull. Amer. Meteorol. Society (BAMS).

In my opinion, the manuscript in the current form is quite bulky, poorly composed, has repetitions, rough, and not easy to follow. There are too many abbreviations in the paper. It sometimes is overwhelming for the readers. Some are not necessary, please refrain from using abbreviations unless unavoidable. Apparently, this is due to the fact that the manuscript is a compilation of different parts prepared by different contributors. Because their result implies something important, I don't think the paper should be rejected. I recommend the authors try to improve the paper in a major revision. I think that the authors should consider reworking the manuscript completely; the paper should be re-elaborated in depth, reduced in size, and be re-written by one co-author in a coherent manner before re-submission.

Either way, my specific comments for a future version are listed below. This is by no means a complete list.

Page 5, line 2. Degree sign for 70 N should be superscript.

Page 5, line 31. Define 'PP' (twice).

Page 20, lines 5 and 7. Introduce the abbreviation 'ECDA3' on first occurrence.

Page 20, line 28 and page 21, line 17. Abbreviation 'LNG' is not defined.

Page 20, line 30. Replace 'Krai' by 'Region' similar to 'Magadan Region' in this line.

Page 21, lines 2, 4, 5, 7. Replace 'Oblast' and 'Krai' by 'Region' similar to 'Magadan Region' on page 20, line 30.

Page 21, line 21. Replace 'Sakhalin' by 'Sakhalin Island'.

Page 22, line 28. There is no 'Sakhalin Sea' in the nature. Do you mean 'Sakhalin

Gulf'?

Figure 4. I believe that the labels on the map should be in English and not in Russian.

---

## Author Comment (AC1) · 10 Dec 2018

See the supplement.

Please also note the supplement to this comment:
https://www.atmos-chem-phys-discuss.net/acp-2018-524/acp-2018-524-AC1-supplement.pdf
* * *

---

## Author Comment (AC2) · 10 Dec 2018

See the supplement.

Please also note the supplement to this comment:
https://www.atmos-chem-phys-discuss.net/acp-2018-524/acp-2018-524-AC2-supplement.pdf
* * *

---

## Author Response (AR1)

The paper introduces the need to study in depth the marine part of the Eurasian arctic, supplementing the ongoing effort of PEEX, that addresses the terrestrial part. The document is a compilation of recent work (the past two decades) on the area, a list of relevant scientific subjects, including the corresponding observational component, and a final part of different proposition to push the research in the area forward, including novel ways of data production and societal impacts. In general the paper is interesting, but it is too long, the different chapters have a lot of superposition and the list of references is incomplete (cites in the text not in the list) or with errors in the references. The impression is that this is a report of the interested group that has been submitted as a paper to ACP. The authors should reduce the text indicating the main points, including clearly the rationale and the actions to perform. This is why I propose a return of the paper to the authors encouring resubmission of a much shorter document. I think it is easier for the authors than to request Major Revisions providing a very long list of Major Comments. Below I summarize some points according to the ACP review guidelines.

1. Does the paper address relevant scientific questions within the scope of ACP? The subject is correct in introducing the need to perform the study in the marine Eurasian area, but it fails to focus on what is really needed. It does not provide a clear rationale on how to proceed neither what actions are essentially needed.

   We have made a major modification to the manuscript to focus on what actions are really needed and how to proceed. Among others, Section 5 (Discussion: The way forward), has been entirely rewritten.

2. Does the paper present novel concepts, ideas, tools, or data? The paper tries to identify all novel methods of measuring to see if they would be of use for their effort. Nevertheless they do not proceed to analyze how these methods should be implemented in such harsh environment.

   In Section 5 we now provide more concrete plans for the implementation of MA-PEEX. In Appendix 1, we have now made in clearer that the new observation technology discussed has been tested in harsh Arctic conditions.

3. Are substantial conclusions reached? No, other than the experiment is necessary(and I agree with that).

As the paper does not include concrete new results, it is hard to present clear conclusions. We have, however, now more clearly presented our evaluation on what should concretely be done to establish the Marine Arctic component of PEEX (Section 5).

4. Are the scientific methods and assumptions valid and clearly outlined? The methods are well described, the assumptions are very reasonable. However, the second half of the paper dealing with the societal impact is not clear. Sections 5 (Socio-economy) and 6 (The way forward) are long, too wordy and should be drastically reduced or be written in a more clear manner.

We have reduced the length of the socio-economy section (now Section 4) by 30% (from 1501 to 1048 words). Previous Sections 6 (The way forward) and 7. (Discussion) included a total of 3755 words. We have merged them together into a new Section 5 (Discussion: The way forward), which only includes 1704 words. We think that the new Sections 4 and 5 are more clear and concrete than the old text.

5. Are the results sufficient to support the interpretations and conclusions? There are not results in the paper, but there is a very good compilation of the work done, so the authors are in very good position to go for the experiment, but this is more for a research proposal than for a scientific paper.

We agree with the reviewer. The approach is different from a typical scientific paper, because the objective of the manuscript is to describe a new component of PEEX in the PEEX Special Issue of ACP.

6. Is the description of experiments and calculations sufficiently complete and precise to allow their reproduction by fellow scientists (traceability of results)? Not applicable, since it is a review/prospective paper.

We agree.

7. Do the authors give proper credit to related work and clearly indicate their own new/original contribution? Yes, the compilation of previous work is excellent, although the reference list misses a lot of the references in the text and some references are inconsistent between the text and the list.

We have corrected the errors and added the missing references.

8, Does the title clearly reflect the contents of the paper? Yes.

We decided to modify the title, because the previous title would not generate interest among readers who have not heard about PEEX.

9. Does the abstract provide a concise and complete summary? The abstract is good.

Thank you!

10. Is the overall presentation well structured and clear? The structure of the paper is good, but every section is too long and there are too many differences between sections, some are reltively concise and very clear others are more speculative and less precise. It is true that the paper addresses many different subjects and it is a big challenge to make a readable document with such a varied contents. However I am positive that a second try will manage to produce a good paper on MA-PEEX.

We have reduced the length of the paper from Introduction until the end of Section 5 (previously 7) by 36% (from 15512 to 9968 words). Structural changes include removal of previous Section 2, as the core issues of the Arctic climate system are now presented in Section

11. Is the language fluent and precise? In general English is good although there aresome language mistakes, easy to correct. There are some changes of writting style between sections, probably due to different contributors, but the main problem is that not every section is written with the same approach, some are good others are weak.

To make the approach and writing style more homogeneous throughout the manuscript, the lead author has alone prepared the revised manuscript, and then collected final input and comments from co-authors.

12. Are mathematical formulae, symbols, abbreviations, and units correctly defined and used? Not applicable, since no mathematical expressions or formulae are used in the text or in the figures.

13. Should any parts of the paper (text, formulae, figures, tables) be clarified, reduced,combined, or eliminated? As indicated, the paper should be substantially reduced in size and also be written in a more internal coherent manner.

See our responses above: 36% reduction in length and internally more coherent rewriting.

14. Are the number and quality of references appropriate? See point 7 on the errors in the references. The list is extensive but appropiate, it reflects well the scientific work made in the area in the recent times.

See above: we have corrected the errors and added missing references.

15. Is the amount and quality of supplementary material appropriate? No supplementary material that I can see. However if the authors produced a shorter version, perhaps they could derive some of the removed parts to supplementary material.

Thank you for the good suggestion! We have moved parts of the original manuscript to appendices.

Timo Vihma on behalf of all co-authors.

Referee 2

We thank the reviewer for his/her work and constructive comments! The referee's comments are copied below, and our responses are written in red below each comment.

Interactive comment on

"Towards the Marine Arctic Component of the Pan-Eurasian Experiment"

By Timo Vihma et al.

Anonymous Referee #2

The manuscript by Vihma et al. describes a marine Arctic component (MA-PEEX) as supplement to the existing terrestrial/atmospheric the Pan-Eurasian Experiment (PEEX) and coastal the SMEAR (Station Measuring Ecosystem-Atmosphere Relations) concepts. The authors' main selling point is that they are going to investigate the behavior of the physical and biogeochemical processes to improve the ocean component of an Arctic observing system based on the marine meteorological, sea ice, and oceanographic observations for the Arctic Ocean itself. In the same time, the SMEAR concept can be applied in coastal and archipelago stations, such as Tiksi, Cape Baranova, etc. The topic of the study is highly interesting and has potential. I also think that the exercise has an obvious practical value. However, I have concerns about the manuscript.

In my view, this document is more like a proposal or 'letter of intent' rather than usual research paper. Ultimately, this is an editorial decision, but the topic appears more appropriate for another journal than ACP. For example, I used to see such papers in Bull. Amer. Meteorol. Society (BAMS).

We agree that the document is not a usual research paper. BAMS would have indeed been an alternative publication forum, but we selected ACP because of its PEEX special issue.

In my opinion, the manuscript in the current form is quite bulky, poorly composed, has repetitions, rough, and not easy to follow. There are too many abbreviations in the paper. It sometimes is overwhelming for the readers. Some are not necessary, please refrain from using abbreviations unless unavoidable. Apparently, this is due to the fact that the manuscript is a compilation of different parts prepared by different contributors.

Because their result implies something important, I don't think the paper should be rejected. I recommend the authors try to improve the paper in a major revision. I think that the authors should consider reworking the manuscript completely; the paper should be re-elaborated in depth, reduced in size, and be re-written by one co-author in a coherent manner before re-submission.

We have reduced the length of the paper by 36% (from Introduction until the end of present Section 5), and strongly reduced the use of abbreviations. We have also removed a lot of repetition and, to make the approach and writing style more coherent throughout the document, the lead author has alone prepared the revised manuscript, and then collected final input and comments from co-authors. Structural changes include removal of previous Section 2, as the core issues of the Arctic climate system are now presented in Section 1 (Introduction) and in the present Section 2 (Existing observations and processes to be studied). Further, a lot of

detailed information on new observation methods and community-based observations have been moved to appendices, and the previous Sections 6 (Way forward) and 7 (Discussion) have been merged together into the new Section 5 (Discussion: The way forward) to make the message more clear and compact.

Either way, my specific comments for a future version are listed below. This is by no means a complete list.

Page 5, line 2. Degree sign for 70 N should be superscript.

Corrected.

Page 5, line 31. Define 'PP' (twice).

We no more use the abbreviation, but write it out (primary production).

Page 20, lines 5 and 7. Introdce the abbreviation 'ECDA3' on first occurrence.

Defined.

Page 20, line 28 and page 21, line 17. Abbreviation 'LNG' is not defined.

Defined.

Page 20, line 30. Replace 'Krai' by 'Region' similar to 'Magadan Region' in this line.

Replaced.

Page 21, lines 2, 4, 5, 7. Replace 'Oblast' and 'Krai' by 'Region' similar to 'Magadan Region' on page 20, line 30.

Replaced.

Page 21, line 21. Replace 'Sakhalin' by 'Sakhalin Island'.

Sentence removed.

Page 22, line 28. There is no 'Sakhalin Sea' in the nature. Do you mean 'Sakhalin Gulf'?

Yes, we meant, but the sentence is now removed to shorten the text.

Figure 4. I believe that the labels on the map should be in English and not in Russian.

We have removed the previous Figure 4. (There is a new Figure 7, with labels in English).

Timo Vihma on behalf of all co-authors.